# Deciphering European Sea Bass (*Dicentrarchus labrax*) Resistance to Nervous Necrosis Virus by Transcriptome Analysis from Early Infection Towards Establishment of Virus Carrier State

**DOI:** 10.3390/ijms26189220

**Published:** 2025-09-21

**Authors:** Dimitra K. Toubanaki, Odysseas-Panagiotis Tzortzatos, Antonia Efstathiou, Vasileios Bakopoulos, Evdokia Karagouni

**Affiliations:** 1Immunology of Infection Group, Department of Microbiology, Hellenic Pasteur Institute, 11521 Athens, Greece; dtouban@pasteur.gr (D.K.T.); ptzortzatos@pasteur.gr (O.-P.T.); toniaef@pasteur.gr (A.E.); 2Department of Marine Sciences, School of The Environment, University of the Aegean, University Hill, Lesvos, 81100 Mytilene, Greece; v.bakopoulos@aegean.gr

**Keywords:** nervous necrosis virus, nodavirus, disease resistance, sea bass, *Dicentrarchus labrax*, head kidney, immune response, transcriptome analysis, RNA-seq

## Abstract

Viral nervous necrosis, caused by the nervous necrosis virus (NNV), is an important threat to aquaculture, causing great economic losses and a high environmental burden. European sea bass (*Dicentrarchus labrax*) is highly affected by NNV, and selective breeding programs for disease resistance have been established in order to achieve a sustainable aquaculture and minimize the need for vaccines, drugs and antibiotics. Resistant and susceptible European sea bass were experimentally challenged with NNV and their head kidney transcriptomes were analyzed at three time points, i.e., 3 hpi, 2 dpi and 14 dpi. Numerous differentially expressed genes (DEGs) were identified in the head kidneys of resistant and susceptible infected vs. non-infected sea bass. Gene ontology enrichment, pathway, and protein–protein interaction analyses revealed that the NNV-resistant fish control their response to viral infection more efficiently, utilizing different mechanisms compared to the susceptible fish. Resistant fish displayed higher levels of interferon-related elements, cytokines, antigen presentation, T-cell activity, apoptosis, and programmed cell death combined with a controlled inflammatory response and more active proteasome and lysosome functions. The susceptible fish appeared to have high immune responses at the early infection stages, accompanied by high expressions of inflammatory, complement and coagulation pathways. Insulin metabolism was better regulated in the resistant fish and the control of lipid metabolism was less effective in the susceptible family. The cytoskeleton- and cell adhesion-related pathways were mostly down-regulated in the susceptible fish, and the intracellular transport and motor proteins were utilized more efficiently by the resistant fish. The present study represents a thorough transcriptomic analysis of NNV infection effects on a resistant and a susceptible European sea bass head kidney. The obtained results provide valuable information on the mechanisms that offers pathogen resistance to a host, with many aspects that can be exploited to develop more efficient approaches to fighting viral diseases in aquaculture.

## 1. Introduction

Aquaculture plays an important role in the global food system, and according to the FAO’s Blue Transformation strategic vision, it should be sustainable, resilient, and minimize environmental impacts, and we should improve biosecurity and disease control with the support of technology and innovation [1]. Genetic improvement for disease resistance through breeding is a sustainable and profitable alternative for protection against pathogens compared to vaccination and drug treatments, and may overcome technical, logistic and cost-related problems, as well as drug toxicity to ecosystems and antimicrobial resistance [2,3,4]. Selective breeding programs in combination with –omics technologies, which have resulted in high-resolution genetic maps, full genome sequences, and transcriptome profiles for several fish species, have provided evidence of the action of specific genetic markers as key components of fish disease resistance mechanisms [5]. Diseases of viral etiology are posing an important threat in aquaculture, causing great economic losses and high environmental burdens [6].

One such virus is nervous necrosis virus (NNV) or nodavirus, which is the causative agent of viral nervous necrosis (VNN) or viral encephalopathy and retinopathy (VER) disease. VNN is responsible for high mortality and morbidity in more than 120 different species from marine and freshwater environments, and it destroys the fish nervous system (e.g., brain, retina, spinal cord). Its clinical symptoms include abnormal swimming behavior, loss of appetite, swim bladder hyperinflation, coloration abnormalities and hemorrhaging brain, which eventually result in host death [2,7,8]. NNV is a member of the Nodaviridae family and belongs to the genus Betanodavirus. The virus consists of a single-stranded RNA genome, with two positive-sense molecules (RNA1 and RNA2) inside a non-enveloped icosahedral capsid. RNA1 encodes an RNA-dependent RNA polymerase (RdRp) with a mitochondrial localization-targeting signal, and RNA2 encodes a capsid protein that can also induce cell apoptosis through the mitochondria-mediated cell death pathway. RNA1 also encodes a sub-genomic RNA3, which is translated into 2 proteins, i.e., B1, which inhibits cell death after virus infection, and B2, which suppresses host short interfering RNA (siRNA) to avoid cleavage and thus allow replication [4]. NNV can be classified into four genotypes based on the T4 region of RNA2: red-spotted grouper nervous necrosis virus (RGNNV), striped jack nervous necrosis virus (SJNNV), tiger puffer nervous necrosis virus (TPNNV) and barfin flounder nervous necrosis virus (BFNNV) [9].

Over the past two decades, genetic studies on NNV resistance in various fish hosts, including Atlantic cod [10], Asian seabass [11], European sea bass [3], gilthead sea bream [12] and leopard coral grouper [13], have suggested that there are specific genetic elements that control viral resistance. The correlation of NNV resistance with specific genes resulting from quantitative trait loci (QTL) studies has indicated the receptor transporting protein 3 (*rtp3*) [14], the growth factor receptor bound protein 2-associated-binding protein 3 (*gab3*) [15], the ribonucleoside-diphosphate M1 (*rrm1*) [16], the T-box transcription factor 21 (*tbx21*) [17] as specific NNV resistance genes in Asian sea bass. However, NNV resistance is a quantitative trait and is controlled by many factors, including gene expression variations, DNA methylation, histone modification, and environmental factors and their interactions [4]. In an attempt to elucidate resistance mechanisms to nodavirus infection, the gilthead sea bream and sea bass transcriptomes, which correspond to resistant asymptomatic carriers and highly virus-susceptible hosts, respectively, were compared [18]. Using an alternative approach, asymptomatic and diseased *Epinephelus moara* naturally infected with nervous necrosis virus were analyzed by RNA-seq and screened molecular markers associated with NNV resistance [19]. In order to derive a more accurate view of disease resistance mechanisms, the use of genetically characterized pathogen-resistant and susceptible hosts for transcriptome analysis has been the subject of increasing investigation over the last decade [20,21,22,23,24,25]. Recently, our group performed transcriptome analyses on the head kidneys of resistant and susceptible European sea bass families following the NNV re-infection of experimental challenge survivors [26].

European sea bass (*Dicentrarchus labrax*) is an important fish in European and Mediterranean aquaculture, with great economic and ecological value [27]. In 2023, the global production of farmed sea bass was estimated at 286,967.69 tonnes [28]; however, infectious diseases pose a significant threat to industry sustainability and the farmed sea bass’ welfare. It is estimated that VNN is currently responsible for 15% of total on-farm infectious disease-related mortalities in European sea bass, and even the surviving fish experience delayed growth, causing economic losses [29]. The *D. labrax* genome has been characterized [30] and its response to NNV infection has been assessed by transcriptome analysis [26,31,32,33,34,35,36,37].

In the present study, two genetically distinct families, i.e., one NNV-resistant and one NNV-susceptible European sea bass family, were subjected to head kidney transcriptome analysis following an experimental NNV infection at various time points (3 h post infection (hpi), 2 days post infection (dpi) and 14 dpi). The transcriptome analysis was focused on the comparison of hosts’ responses to NNV infection. More specifically, the responses of NNV-resistant fish and NNV-susceptible fish were compared to the respective uninfected controls, in order to decipher the molecular mechanisms used by each family to overcome the infection. The differences in the two families’ responses indicate the essential elements that result in disease resistance and provide information regarding the NNV resistance traits, which are not covered solely by genetic analysis. Moreover, the analysis of the time point (14 dpi) at which fish appear to act as asymptomatic carriers also reveals specific host mechanisms that could be key parts of disease resistance. The present study was focused on the fish head kidney, which plays a crucial role in teleosts immunity [38], providing insights into NNV resistance mechanisms at a systemic level. The resistant and susceptible families’ responses were assessed in three complementary ways—gene ontology (GO) enrichment, pathway (KEGG), and protein–protein interaction (PPI) analyses—which revealed key genes and pathways important to disease resistance. Overall, the analysis of genetic factors and the molecular interplay that is involved in pathogen resistance mechanisms could result in the production of more resilient fish populations, but also could improve genetic-selective breeding strategies using specific genes or proteins’ levels as biomarkers. Also, such studies could lead to the development of more effective disease treatments, contributing to sustainable aquaculture.

## 2. Results

### 2.1. Experimental Challenge of NNV-Resistant and -Susceptible D. labrax Families

Experimental NNV challenge via intramuscular injection into *D. labrax* fish of resistant (R) and susceptible (S) families was utilized to study NNV infection progress for a 28-day period post infection. A summary of the experimental setup is given in Figure 1. It should be noted that throughout the manuscript, R is used as the symbol of the resistant family and S as the symbol of the susceptible family. The letter I following R or S is used for the infected groups and NI is used for the non-infected groups. Therefore, RI stands for resistant infected fish, RNI symbolizes the resistant non-infected fish, SI stands for susceptible infected fish and SNI symbolizes the susceptible non-infected fish.

The mortality rates were recorded daily for both families throughout the study. In NNV-resistant fish (R), typical signs of VNN were observed from day 3 and the first death occurred on day 4, while the NNV-susceptible fish (S) had apparent VNN symptoms and the first death occurred on day 2. The cumulative survival rates for each family have been previously reported in Toubanaki et al. [26]. The R family showed a decline in survival from day 4 to day 15 post infection (86.27% survival), which had ceased by the end of the experiment. On the contrary, the S family showed a sharp decline in fish survival from day 2 to day 11 post infection, with maximum mortality on day 5 (50.46% survival). No mortalities were registered in non-infected sea bass, whose survival remained 100% up to the end of the experiment.

The sea bass brain was tested for viral NNV RNA1 presence at three time points, i.e., 3 hpi, 2 dpi and 14 dpi (Figure 2). At both 3 hpi and 2 dpi we saw no detectable levels of the virus in fish brain for both families, with the exception of one fish belonging to the resistant family. The viral load of the surviving fish 14 dpi was 20.6 ± 9.8 × 10^11^ TCID_50_ of the NNV per µg of total RNA for the R family and 29.7 ± 14.3 × 10^11^ TCID_50_ of the NNV per µg of total RNA for the S family. Despite the presence of the virus in the brain, no phenotypic or behavioral changes were observed in infected fish 14 dpi. No viral load was observed in the non-infected groups.

### 2.2. Summary and Assessment of NNV-Challenged D. labrax Head Kidney RNA Sequencing (RNA-Seq) Data

The head kidney total RNA, isolated from both resistant and susceptible NNV-infected (*n* = 26) and non-infected (*n* = 19) fish at three different time points (3 hpi, 2 dpi, 14 dpi), was subjected to RNA-seq. The sequencing of the 45 libraries yielded a total of 466,644,296 and 256,948,756 sequences for NNV-infected and non-infected head-kidney samples, respectively (Appendix A). Raw data reads are available in the NCBI database (Accession No. PRJNA1030357). All reads were aligned to the reference genome at a high rate (66.5–97.8%).

### 2.3. Differential Expression Analysis Between Infected and Non-Infected Groups

Differentially expressed transcripts were analyzed using the Bioconductor package DESeq2 (version 3.12), and were identified according to log_2_FC ≥ |2| and an adjusted *p*-value (FDR) threshold less than 0.05. At each time point, the transcriptome data were compared among four groups: the resistant infected fish (RI), the susceptible infected fish (SI), the resistant non-infected fish (RNI) and the susceptible non-infected fish (SNI). At the 3 hpi time point, the pairwise comparison of RI vs. RNI identified 1559 genes significantly up-regulated and 387 genes significantly down-regulated, while SI vs. SNI resulted in 13 genes significantly up-regulated and 160 genes significantly down-regulated. Two days post infection (2 dpi), the numbers of up- and down-regulated genes were 32 and 63, respectively, for RI vs. RNI. Similarly, the SI vs. SNI comparison identified 64 genes significantly up-regulated and 175 genes significantly down-regulated. When the RI samples were compared with RNI samples at 14 dpi, it was found that no gene was deregulated. Finally, the SI vs. SNI comparison revealed the significant up- and down-regulation of 9 and 13 genes, respectively (Appendix A). Pairwise comparisons were also performed between the resistant and susceptible families at each time point. At 3 hpi, the RI vs. SI comparison resulted in 45 genes significantly up-regulated and 38 genes significantly down-regulated. The RNI vs. SNI comparison identified 120 genes significantly up-regulated and 346 genes significantly down-regulated. The up- and down-regulated genes for RI vs. SI were 50 and 45, respectively, at 2 dpi. Similarly, the RNI vs. SNI comparison identified 285 genes significantly up-regulated and 499 genes significantly down-regulated. When the RI samples were compared with SI samples at 14 dpi, it was found that 77 genes were up-regulated and 39 were down-regulated, while the RNI vs. SNI comparison revealed the significant up- and down-regulation of 37 and 38 genes, respectively.

The up- and down-regulated DEGs corresponding to the described comparisons at each time-point are depicted in Figure 3. A striking pattern of global gene expression differences between the families was evident, in particular at the early time points (3 hpi and 2 dpi). The resistant family showed higher numbers of both up-regulated and down-regulated transcripts than the susceptible family at the initial time point (3 hpi), which was reversed at the next time-point (2 dpi). At 14 dpi, gene expression deregulation was kept at low levels, but was still higher in the susceptible fish. The common up- and down-regulated genes in different time points were low for both families. In detail, at 3 hpi and 2 dpi, 7 and 11 genes were commonly up- and down-regulated, respectively, in the resistant family, while 4 and 40 genes were commonly up- and down-regulated in the susceptible family at those time points. No genes were found to be significantly deregulated between 2 and 14 dpi for both families. The number of common de-regulated DEGs between the resistant and the susceptible family head kidney response (i.e., RI vs. RNI and SI vs. SNI) was relatively low (<30 common genes) at all time points, reflecting a totally different head kidney response between the two families. A differential gene expression clustering analysis indicated high intragroup similarity and intergroup differences between all compared groups (Appendix A).

### 2.4. GO Classification and Enrichment Analysis Between Infected and Non-Infected Experimental Groups of the NNV-Resistant (R) and Susceptible (S) Families

In order to identify the potential molecular mechanisms underlying VNN disease resistance, we followed a three-step strategy, which consisted of the identification of DEGs resulting from the comparison of infected versus non-infected groups for each family at each time point and their GO classification; the identification of the family-specific GOs of the up- and down-regulated genes for the resistant and susceptible families at each time point, separately; the identification of family-specific GOs at all time points.

The functional enrichment analyses (adj. *p* < 0.05) between the resistant infected (RI) and non-infected NNV (RNI) experimental groups at the 3 hpi, 2 and 14 dpi time points are listed in Appendix A, and the top 20 GOs are shown in Figure 4. For the up-regulated DEGs (Figure 4A), the most highly enriched category at all time points was the biological process (3 hpi, 60/104; 2 dpi, 17/37; 14 dpi, 11/20), followed by the molecular function (3 hpi, 27/104; 2 dpi, 12/37; 14 dpi, 5/20) and the cellular component (3 hpi, 17/104; 2 dpi, 8/37; 14 dpi, 4/20) categories. For the down-regulated DEGs, the GOs resulting from enrichment analysis are listed in Appendix A and the top 20 GOs are shown in Figure 4B. The most enriched categories at 3 hpi and 2 dpi were biological process (3 hpi, 47/81; 2 dpi, 28/54), followed by the molecular function (3 hpi, 19/81; 2 dpi, 15/54) and the cellular component (3 hpi, 15/81; 2 dpi, 11/54) categories. The late time point (14 dpi) appeared to show GO enrichment only in the molecular function (1/2) and the cellular component (1/2) categories.

The most enriched categories for the three time points, for both the up- and down-regulated genes (except down-regulated genes’ GOs at 14 dpi) were (1) immune system process, signaling, anatomical structure development, cell differentiation, and regulation of DNA-templated transcription (BP category); and (2) extracellular space and nucleoplasm (CC category). The GO terms uniquely enriched for the up-regulated genes at 3 hpi numbered 64 and include protein-containing complex assembly, transmembrane transport, cell adhesion, endoplasmic reticulum, Golgi apparatus, transporter activity, membrane organization, protein catabolic process, molecular transducer activity, lipid metabolic process, and nervous system process. The GO terms uniquely enriched for the up-regulated genes at 2 dpi, numbered five, i.e., ion binding, chromosome, organo-nitrogen compound metabolic process, organic cyclic compound binding and cytoplasmic vesicle, whereas only one term was uniquely enriched in up-regulated genes at 14 dpi (carbohydrate derivative metabolic process). The terms uniquely enriched for the down-regulated genes at the 3 hpi time point numbered 32, including RNA binding, mRNA metabolic process, wound healing, circulatory system process, extracellular matrix, defense response to other organism, mitochondrion, endosome, translation regulator activity and nucleolus; the GO terms uniquely enriched for the down-regulated genes at 2 and 14 dpi numbered six (cytoplasmic vesicle, cytoskeleton, small molecule metabolic process, DNA recombination, catalytic activity acting on DNA, cytoskeletal motor activity) and one (organelle), respectively.

The results of the functional enrichment analyses (adj. *p* < 0.05) between the susceptible infected (SI) and non-infected NNV (SNI) experimental groups at the 3 hpi, 2 and 14 dpi time points are listed in Appendix A, and the top 20 GOs are shown in Figure 5. For the up-regulated DEGs (Figure 5A), the most enriched category at all time points was the biological process (3 hpi, 7/15; 2 dpi, 13/32; 14 dpi, 7/12), followed by the cellular component (3 hpi, 5/15; 2 dpi, 9/32; 14 dpi, 3/12) and the molecular function (3 hpi, 3/15; 2 dpi, 10/32; 14 dpi, 2/12) categories. For the down-regulated DEGs, the GOs resulting from enrichment analysis are listed in Appendix A, and the top 20 GOs are shown in Figure 5B. The most enriched category in all cases was the biological process (3 hpi, 41/75; 2 dpi, 43/74; 14 dpi, 17/31), followed by the molecular function (3 hpi, 20/75; 2 dpi, 16/74; 14 dpi, 7/31) and the cellular component (3 hpi, 14/75; 2 dpi, 15/74; 14 dpi, 7/31) categories.

The only enriched category at all time points for both the up- and down-regulated DEGs was the anatomical structure development (BP) category. The GO terms uniquely enriched for the up-regulated genes at 3 hpi were hydrolase activity, extracellular region, nucleolus, protein binding, primary metabolic process, cellular component biogenesis, and organic cyclic compound binding and transport. The GO terms uniquely enriched for the up-regulated genes at 2 dpi numbered 19, including GTP binding, carbohydrate derivative metabolic process, nucleobase containing small molecule metabolic process, extracellular space, lipid droplet, ATP-dependent activity, endoplasmic reticulum, transferase and oxidoreductase activity, RNA metabolic process, programmed cell death and RNA binding, whereas 4 terms were uniquely enriched for the up-regulated genes at 14 dpi, i.e., membrane organization, nucleus, mitochondrion organization and cytoskeletal protein binding. The terms uniquely enriched for the down-regulated genes at the 3 hpi time point numbered 15, including ATP-dependent activity, cytoskeletal motor activity, nuclear envelope, receptor ligand activity, cellular modified amino acid metabolic process, cytokinesis, microtubule organizing center, catalytic activity, acting on DNA and syntaxin binding. The GO terms uniquely enriched for the down-regulated genes at 2 dpi numbered 13, including molecular function regulator activity, chromatin organization, peroxisome, nucleolus, DNA repair, vitamin metabolic process, protein tag activity, endocrine process, peroxisome organization and regulatory ncRNA mediated gene silencing. Finally, the GO terms uniquely enriched for the down-regulated genes at 14 dpi were nucleus, endomembrane system, response to stress and hydrolase activity.

In order to further understand the expression patterns of the resistant and susceptible families at the studied time points, the GO enrichment analyses’ results for each family were compared to identify the family-specific GOs of the up- and down-regulated genes for the resistant and susceptible families at each time point. For the up-regulated genes at the 3 hpi time point, both families had seven commonly enriched GO categories, including the immune system process, plasma membrane and anatomical structure development. The resistant uniquely enriched GOs numbered 97, and the top categories were signaling, cell differentiation, transferase activity, regulation of DNA templated transcription, vesicle-mediated transport, programmed cell death, mitochondrion, defense response to other organism, nervous system process, autophagy, mitochondrion organization, RNA binding, and inflammatory response. The susceptible uniquely enriched GOs were numbered eight, and were hydrolase activity, extracellular region, protein binding, primary metabolic process, cellular component biogenesis, organic cyclic compound binding, transport, macromolecule metabolic process. Respectively, both families had 61 commonly enriched GO categories for the down-regulated genes at 3 hpi. The resistant uniquely enriched GOs numbered 20, including RNA binding, the mRNA metabolic process, regulatory ncRNA-mediated gene silencing, GTP binding, autophagy and catalytic activity, acting on RNA categories. The susceptible uniquely enriched GOs numbered 14, including the generation of precursor metabolites and energy, cytoskeletal motor activity, mitotic cell cycle, protein maturation, DNA recombination and cytokinesis categories.

At the 2 dpi time point, the up-regulated genes had 14 commonly enriched GO categories for both families, including the immune system process, plasma membrane and anatomical structure development, as before. The resistant uniquely enriched GOs numbered 23, and were cytoskeletal motor activity, cytoskeletal protein binding, cytoskeleton organization, microtubule organizing center, chromosome segregation, microtubule-based movement, muscle system process, structural molecule activity, vesicle-mediated transport, cytoplasmic vesicle and cell motility, among others. The susceptible uniquely enriched GOs numbered 18, including GTP binding, carbohydrate derivative metabolic process, defense response to other organism, GTPase activity, lipid droplet, mitochondrion, RNA metabolic process and RNA binding. For the down-regulated genes at 2 dpi, both families had 43 commonly enriched GO categories. The resistant uniquely enriched GOs numbered 11 (cytoplasmic vesicle, receptor ligand activity, cilium, muscle system process, nuclear envelope, small molecule metabolic process, nuclear chromosome, mitotic nuclear division, catalytic activity, acting on DNA, ATP-dependent activity, cytoskeletal motor activity). The susceptible uniquely enriched GOs numbered 31, including defense response to other organism, molecular function regulator activity, wound healing, protein maturation, mitochondrion, lysosome, peroxisome, regulatory ncRNA mediated gene silencing and ribosome.

At the late time point, i.e., 14 dpi, the up-regulated genes had five commonly enriched GO categories for both families (defense response to other organism, signaling, anatomical structure development, mitochondrion and cell differentiation). The resistant uniquely enriched GOs numbered 15, including immune system process, extracellular space, transcription regulator activity, mRNA metabolic process, catalytic activity, acting on RNA, DNA binding and hydrolase activity. The susceptible uniquely enriched GOs numbered seven, and were membrane organization, nucleus, mitochondrion organization, GTPase activity, protein containing complex assembly, cytoskeletal protein binding and cytoskeleton. For the down-regulated genes, no commonly enriched GOs of both families were found at 14 dpi. The resistant uniquely enriched GOs numbered 2 (oxidoreductase activity and organelle), whereas the susceptible uniquely enriched GOs numbered 31, including plasma membrane, signaling, cytoskeleton, anatomical structure development, cytoskeleton organization, cytosol, muscle system process, cytoskeletal protein binding, immune system process, nervous system process and programmed cell death, among others.

### 2.5. Pathway Analysis Between Infected and Non-Infected Experimental Groups of the NNV-Resistant (R) and -Susceptible (S) Families

Pathway analysis was performed for NNV-infected versus non-infected experimental groups for both families, i.e., resistant infected (RI) vs. non-infected (RNI) and susceptible infected (SI) vs. SNI. All annotated transcripts were run against the KEGG database and the enriched pathways were determined by Gene Set Enrichment Analysis (GSEA). The enriched pathway categories shown by the RI vs. RNI analysis of the experimental groups at the 3 hpi, 2 and 14 dpi time points are summarized in Table 1 and analyzed in Appendix A, and the top 30 pathways are shown in Figure 6.

The pathways that were enriched in all studied time-points are involved in the following: (1) immune system—cytosolic DNA-sensing, NOD-like, C-type lectin and RIG-I-like receptor signaling, antigen processing and presentation, and platelet activation; (2) cellular processes—necroptosis and motor proteins; (3) signal transduction—calcium, AMP-activated protein kinase (AMPK), cAMP and sphingolipid signaling; (4) metabolism—arginine and proline metabolism, fructose and mannose metabolism; (5) genetic information processing—replication and repair (nucleotide excision repair, Fanconi anemia), transcription (RNA polymerase, spliceosome) and translation (ribosome); (6) viral infectious diseases (influenza A, COVID-19), and bacterial infectious diseases (pertussis, Yersinia, Salmonella and pathogenic *E. coli* infections, shigellosis).

The early time point (i.e., 3 hpi) shared the majority of enriched pathways with the 2 dpi time point, namely infectious, bacterial, immune, cardiovascular, endocrine and metabolic diseases. Interestingly, parasitic infectious diseases-related pathways were mostly enriched at 2 dpi, and the neurodegenerative diseases pathways were enriched at 14 dpi. Metabolism seemed to be mostly dependent on carbohydrates and glycan at the early stage of infection (3 hpi), while amino acid metabolism seemed to be utilized at 2 dpi. At the 14 dpi time point, the metabolism of amino acids, including glutathione and oxidative phosphorylation, seemed to provide the main energy sources. The nervous system-related pathways seemed to be more active at 3 hpi, while the immune system related pathways were more active (in terms of implicated pathways) at 2 dpi. Finally, signal transduction was more intense at the 3 hpi and 2 dpi time points.

The results enriched pathway categories analysis between the SI vs. SNI experimental groups at the 3 hpi, 2 and 14 dpi time points are summarized in Table 2 and analyzed in Appendix A, and the top 30 pathways are shown in Figure 7.

The pathways enriched in the NNV-susceptible family at all time-points were (1) immune system—cytosolic DNA-sensing, NOD-like and C-type lectin receptor signaling; (2) cellular processes—necroptosis; (3) genetic information processing—translation (ribosome); (4) viral infectious diseases (influenza A, COVID-19), and bacterial infectious diseases (Yersinia, Salmonella and pathogenic *E. coli* and *Staphylococcus aureus* infections).

In the susceptible family, the 3 hpi time point shared the majority of enriched pathways with the 14 dpi time point of neurodegenerative and cardiovascular diseases, while the 2 and 14 dpi time points showed similar profiles of infectious and bacterial diseases. Parasitic infectious diseases-related pathways were mostly enriched at the initial time point, in contrast with the resistant family. In this case, metabolism was mostly dependent on carbohydrates, nucleotide and energy metabolism (oxidative phosphorylation) at the 3 hpi and the 14 dpi time points, while glucan and lipid metabolism seemed to be utilized at 2 dpi. Interestingly, in this family, the pathways implicated in xenobiotics biodegradation and metabolism were highly enriched at 3 hpi. Both the nervous system- and immune system-related pathways seemed to be more active at 3 hpi. Finally, signal transduction was strikingly intense at the early time point.

### 2.6. Protein–Protein Interaction (PPI) Analysis Between Infected and Non-Infected Experimental Groups of the NNV-Resistant (R) and -Susceptible (S) Families

To gain further insights into the disease resistance mechanisms, the zebrafish orthologs of all differentially expressed genes between the NNV-infected and -non-infected experimental groups of both resistant and susceptible families were manually determined for the three time points. All initially found DEGs were manually curated to find zebrafish homologs, and the resulting gene names were analyzed by the Search Tool for the Retrieval of Interacting Genes/Proteins database of the STRING database. The initial DEGs and those that were finally analyzed for all conditions are summarized in Table 3.

PPI analysis with k-means clustering resulted in a dense cluster network for the resistant family at the 3 hpi time point, and distinct clusters for the resistant family at 2 dpi and the susceptible family at all time points (Figure 8). The top 10 hub genes identified in the PPI network in each case are presented in Table 4 and Table 5. The resistant family hub genes at the 3 hpi time point include Golgi SNAP receptor complex member 2 (*gosr2*), CD59 glycoprotein, transport and trafficking proteins (*sec23a, sec22c*), and coagulation-related proteins (*serpina1, mcfd2*), while the susceptible family hub genes at the same time point include myosins (*muh7, mylz3*), troponins (*tnnt2a, tnni2a.4, tnnt3a, tnni1a*) and actins (*actn3a, actc1a*). At the following time point (2 dpi), the resistant family hub genes contained several kinesin protein members (*kif20a, kif5aa, kif1aa*), the nuclear receptor subfamily 4 group A (*nr4a1*), dual specificity protein phosphatase 1 (*dusp1*) and interleukin-1 beta (*il1b*), and the susceptible family hub genes contained alpha-2-HS-glycoprotein (*ahsg1*), plasminogen and anti-plasmin (*plg, serpinf2a*), aminotransferases (*agxtb, agxta*), heparin and complement components (*serpind1, c8a, c5*). Finally, at the 14 dpi time point, the susceptible family hub genes included glyceraldehyde-3-phosphate dehydrogenase and aldolase a (*gapdh, aldob*), actins (*actc1b, actn2b, actb1*) and ryanodine receptor 1 (*ryr2b*).

### 2.7. qRT-PCR Assay of Selected Genes in the Head-Kidney of NNV-Resistant and NNV-Susceptible D. labrax Families

The expression levels of 10 selected genes were analyzed by quantitative RT-PCR to validate the DEGs identified by RNA-seq. The analyzed genes were the fructose-bisphosphate aldolase a (*aldoaa*), beta-galactosyltransferase 1-like (*b4galt1*), claudin-5-like (*cldn5b*), immunoglobulin light chain (*iglc*), protein nlrc3-like (*nlrc3l*), lrr and pyd domains-containing protein 12-like (*nlrp12*), protein phosphatase mitochondrial-like (*ppm1k*), troponin fast (*tnni2a.1*), novel protein titin (*ttnh*) and zinc finger mym-type protein 1-like (*zmym1*), which were highly expressed by the resistant and/or the susceptible family. As shown in Figure 9, the expressions of the genes when the infected fish were compared with non- infected fish agree with the expression results obtained from the transcriptome analysis.

## 3. Discussion

Fish nervous necrosis virus is an important viral pathogen affecting a wide range of fish hosts worldwide [8]. Especially in the Mediterranean, NNV is a constant threat to European sea bass (*D. labrax*), which plays a key role in aquaculture with significant implications for feeding the human population and environmental pollution, as well as impacts on local and European economies. Herein, the transcriptomic differences between two sea bass families with divergent resistance to NNV infection were analyzed. Transcriptome profiles were assessed 3 hpi, 2 dpi and 14 dpi for infected and non-infected experimental groups, and the infection responses of each family were studied. The viral infection affected only a small number of common genes in both families, at all time points, suggesting that the influence of the host’s genetic background on infection response dictates the specific nature of gene expression modulation, as previously reported for bacterial infection [24].

Three time points were selected for analysis, in an attempt to gain a better understanding of the viral infection dynamics in resistant and susceptible sea bass. The time points were selected based on our previous study [36], where it was found that immune gene modulation was evident at 3 hpi, which was designated as the early infection time point. The 2 dpi (48 hpi) time point was selected as the second time point, since at that point the host immune response seems to reach a stabilized level of gene expression. The final time point was set at 14 dpi, since that was found to be a key time point for the establishment of a virus carrier state in the host [36]. At the initial time point, no viral load was detected in both families. Two days later (2 dpi), only one fish belonging to the resistant sea bass group had a detectable viral load, even though the clinical signs had appeared and a death had occurred in the susceptible fish. At the 14 dpi time point, both families showed a considerable amount of replicated virus in the brain, which is the virus’ target organ, with similar levels, but the fish did not show any clinical signs of the disease. This observation is in agreement with previous independent studies [22], confirming that virus-resistant fish are actively infected; therefore, genetic resistance mechanism cannot be entirely based on the virus’ inability to infect the host.

Even though VNN infection is neurotropic and the virus is mainly localized in the brain [8], fish head kidney plays a crucial role in immune responses following a viral infection, since it is one of the most important lymphoid tissues in teleosts [38]. As a result, the systemic host responses, which manifest in the head kidney, play a key role in disease resistance mechanisms’ elucidation. To this end, numerous differentially expressed genes (DEGs) were identified in the head kidneys of resistant infected vs. non-infected (RI vs. RNI) and susceptible infected vs. non-infected (SI vs. SNI) sea bass experimental groups, at various time points. The resistant and susceptible families’ responses were studied utilizing complementary approaches, i.e., gene ontology (GO) enrichment, pathway (KEGG) enrichment, and protein–protein interaction (PPI) analysis, in order to examine expression patterns and obtain a more comprehensive understanding of the biological functions implicated in each sea bass family’s responses to NNV infection. Based on the constructed PPI networks, several hub genes for each family at each time point were identified, and their potential effect on viral resistance has also been discussed.

The host’s response to a viral infection reflects a complex mechanism, with multiple components and many levels of reaction to the intruder. The optimum activation and interaction of these components in relation to the well-being of the host, and the deterrence of the virus’ armory by which it exploits the host’s machinery for its own benefit, determine the host’s resistance or susceptibility to a pathogen. The main player affecting host response is the immune system, with innate and adaptive arms. When a pathogen enters an organism, a wide range of intracellular pathways are activated, including the complement and coagulation cascades. Host cells use antimicrobial proteins, pathogen-restrictive compartmentalization and cell death in their defense against pathogens, procedures that activate the migration of cells. Therefore, cell mobility, immune cell activation and the destruction of the intruder are heavily based on cytoskeleton function in eliciting an effective response [39].

It is well established that the NNV infection of European sea bass provokes significant immune responses [2,8], which regulate the infection fate and disease severity. Therefore, it can be safely assumed that the observed phenotypic differences between resistant and susceptible fish head kidney responses are heavily dependent on the expression profiles of genes directly involved in the host’s immune response to a pathogen. Indeed, in the present study of GO enrichment, the term “immune system process” was enriched in all time points for both families, with up-regulated and down-regulated transcripts, with the exception of the down-regulated genes for the resistant family and the up-regulated genes for the susceptible family. Apart from sea bass infected with NNV [26], the same trend was observed in infectious pancreatic necrosis virus (IPNV)-infected Atlantic salmon [22], *Aeromonas salmonicida*-infected turbot [24], and Japanese flounder infected with *Edwardsiella tarda* [23].

Even though several immune-related pathways were enriched in both resistant and susceptible fish at all time points (‘cytosolic DNA sensing’, ‘C-type lectin receptor’ and ‘NOD-like receptor’ signaling) or specifically at the 3 hpi and 2 dpi time points, i.e., ‘IL-17 signaling pathway’ and ‘neutrophil extracellular trap formation’, quite different immune system components seem to be involved in each family response, at each phase of the infection. The ‘NF-kappa B signaling pathway’ was only enriched in the resistant family at 3 hpi and 2 dpi. The ‘toll-like receptor’, the ‘toll and Imd’, the ‘TNF’, the ‘JAK-STAT’, the ‘Fc epsilon RI’ and the ‘chemokine’ signaling pathways were enriched more intensively at 3 hpi and 2 dpi in the resistant family, but were only enriched at 3 hpi in the susceptible fish. In a study of IPNV-infected Atlantic salmon fry, the ‘toll-like receptor’, the ‘TNF, the ‘chemokine’ and the ‘Jak-STAT’ signaling pathways were also enriched in both resistant and susceptible families, but at later time points (20 dpi for resistant fish; 7 and 20 dpi for susceptible fish) [22]. The chemokine signaling pathway also appeared to be up-regulated in Japanese flounder, which was resistant to *E. tarda* [23]. The ‘RIG-I-like receptor’ signaling pathway, which is implicated in the activation of interferon stimulated genes (ISGs), was highly enriched in the resistant family, but interestingly it was also enriched at 14 dpi in the susceptible family, in agreement with findings on the IPNV-infected Atlantic salmon fry [22], confirming the important role of interferon response in resistance mechanisms. It is highly impressive that the ‘cytokine–cytokine receptor interaction’ and the ‘viral protein interaction with cytokine and cytokine receptor’ were only enriched in the resistant family, at the 3 hpi and 2 dpi time points. At those time points, the ‘adipocytokine signaling pathway’ was also enriched, but this pathway was also utilized by the susceptible family at 3 hpi. The ‘cytokine–cytokine receptor interaction’ pathway was found to be enriched in IPNV-resistant Atlantic salmon fry at a late infection time point (20 dpi), but in that study, it was more intensively enriched in susceptible samples at 7 and 20 dpi [22]. The ‘cytokine-cytokine receptor interaction signaling pathway’ was also enriched in an *E. tarda*-resistant Japanese flounder compared to the susceptible group [23]. In our previously published study [26], where the European sea bass responses were assessed 7 days post NNV re-infection, it was also found that the resistant family utilized pathways related to ‘cytokine–cytokine receptor interaction’, ‘viral protein interaction with cytokine and cytokine receptor’ and ‘neutrophil extracellular trap formation’. On the other hand, the susceptible family’s response was mostly depended on ‘T cell receptor signaling’, ‘toll-like receptor cascades’, and the ‘regulation of NF-kappa B signaling’, suggesting that the susceptible fish response to re-infection is more similar to the resistant fish response to a primary infection, an interesting observation that requires further study.

Genes involved in antigen recognition and presentation pathways were expressed at higher levels by the resistant family at all time points. In the susceptible fish, antigen recognition and presentation was active at 3 hpi as expected, but it was also enriched at 14 dpi, an observation that is possibly related to carrier state establishment. The related ‘T cell receptor signaling pathway’ was highly enriched at the early time point by both families and remained active at 2 dpi only in the resistant family. A higher expression of numerous genes related to the differentiation and activation of T-cells was also observed in an *A. salmonicida*-resistant turbot family following infection, implying a predisposition to the better initiation of the adaptive immune response in resistant fish [24]. On the other hand, the ‘B cell receptor signaling pathway’ was enriched at 2 and 14 dpi in the resistant fish, while the susceptible family had a different profile, with enhanced B cell signaling as early as 3 hpi and 14 dpi. It is noteworthy that the ‘intestinal immune network for IgA production’ was only enriched in the resistant family at 2 dpi. The ‘Th17 cell differentiation’ was also enriched at the same time point for the resistant family, but only at the 3 hpi time point in the susceptible fish. The ‘antigen processing and presentation’ pathway and the closely associated ‘intestinal immune network for the IgA production pathway’ and ‘T/B cell receptor signaling pathways’ were found significantly enriched in resistant Japanese flounder following *E. tarda* infection [23]. Antigen-presenting cells are also involved in the delivery of antigen peptides from phagosomes to CD4^+^ T cells [40]. In the present study, both ‘phagosome’ and ‘Fc gamma R-mediated phagocytosis’ were enriched at 3 hpi and 2 dpi in the resistant family and at 3 hpi in the susceptible family, indicating a more active process related to adaptive immune responses for the resistant family.

The ‘ubiquitin-mediated proteolysis’ was up-regulated at 3 hpi and 2 dpi in the resistant family and 3 hpi in the susceptible family. In the IPNV-infected Atlantic salmon fry, the ‘ubiquitin-dependent degradation’ was found to be mainly enriched in the susceptible family at 7 dpi [22]. The ubiquitin system plays a critical role in antigen presentation, and our findings support the hypothesis of its contribution to fish immune defense against infection by the suppression of virus production [41]. The ‘proteasome’ pathway was enriched at 3 hpi and 14 dpi in both families, and the related GO term was also enriched in the up- and down-regulated DEGs of the resistant family at 3 hpi and in the down-regulated DEGs of the susceptible family at 3 hpi and 2 dpi. Interestingly, the ‘lysosome’ pathway was enriched only in the susceptible family at 3 hpi and 2 dpi. However, the lysosome GO term was enriched in the up- and down-regulated genes of the resistant family at 3 hpi, and the down regulated genes of the susceptible family at 3 hpi and the 2 dpi. Also, the ‘lysosome organization’ GO term was enriched in the up-regulated genes of the resistant family.

Overall, the observed immune responses to NNV were more diverse in terms of implicated pathways and intense in resistant fish, especially at the 3 hpi and 2 dpi time points, but were still dominant at the 14 dpi time point, enabling the establishment of the virus carrier state in the host. The host responses included almost every major component of the innate immune system, since several genes involved in antigen presentation and T and B cell activation appeared to be up-regulated in fish pathogen-resistant families, suggesting a more effective initiation of the adaptive immune response combined with a controlled inflammatory response. That kind of response has advantages in a virus resistance mechanism, since a controlled inflammatory response provides more advantages in terms of survival than does an exacerbated response [24]. These observations are in agreement with those of the IPNV-infected head kidney analysis of resistant salmon, which showed significantly more highly expressed pro-inflammatory genes and transcription factors [20]. In another study, the resistant Atlantic salmon had a limited and prolonged immune response to IPNV infection, while the susceptible fish had an acute short immune response accompanied by high inflammation [21]. On the contrary, IPNV-resistant Atlantic salmon fry showed a milder immune response, with the up-regulation of M2 macrophage system-related genes, while cytokine genes and inflammatory activity were up-regulated in susceptible fish. That mechanism seemed to be more effective in enabling survival of the infection, at least at the fry stage [22]. Similarly, in salmon anaemia virus (ISAV)-susceptible Atlantic salmon, several innate immunity-related transcripts were up-regulated, but not enough to protect the host, whereas the resistant fish had lower inflammatory responses, which allowed the survival of infected fish until the activation of adaptive immunity pathways, which led to virus clearance [42]. Taking into consideration similar observations for bacterial infections in resistant and susceptible fish [23,24], it can be assumed that the ability of a resistant family to limit the infection’s effects is supported by a controlled immune response, which includes the high expression of interferon-related pathway elements, cytokines, antigen presentation and T-cell activity, combined with a controlled inflammatory response to infection.

Viral pathogenesis is heavily regulated by the complement system, which plays an important role in innate immunity through the recognition of invading pathogens and the subsequent triggering of effector pathways, contributing to viruses’ neutralization and the killing of infected cells. Complement is also involved in adaptive immune system modulation, host inflammatory responses, immunologic memory, and even in tissue regeneration. However, the host complement system can be ‘manipulated’ by the invading virus for its own benefit, with the substitution of the host’s complement regulators by viral homologues viruses. For example, viruses may use complement receptors to enter the cell and facilitate their spread [43]. In fish, it has been reported that complement genes in VHSV-infected rainbow trout are down-regulated, suggesting that this pathway plays a role in viral suppression [44]. During systemic inflammation, complement activation is often accompanied by coagulation cascade activation, which plays a critical role in fighting infections via entrapment and the prevention of the systemic dissemination of pathogens, including viruses [45]. Therefore, coagulation is tightly related to viral infection, playing an important role in sea bass responses following an infection by NNV. In the present study, the complement and coagulation cascades were enriched in the susceptible family at 3 hpi and 2 dpi. Similarly, the coagulation and complement pathway was found to be consistently down-regulated in IPNV-susceptible Atlantic salmon fry at 20 dpi [22]. The KEGG pathway analysis revealed that the ‘hematopoietic cell lineage’ pathway was highly enriched only by the resistance family at 2 dpi, in accordance with a previous observation in *E. tarda*-resistant Japanese flounder [23], a finding worth further evaluation since hematopoietic cell lineage is an important signaling pathway.

In accordance with our previous findings on the NNV re-infection of resistant and susceptible sea bass [26], the ‘necroptosis’ pathway was enriched in both families at all time points. The ‘apoptosis’ pathway was enriched only at the 3 hpi time point in the two families, but remained enriched at 2 dpi only in the resistant fish, while the ‘p53 signaling pathway’ was enriched by the susceptible family at 3 hpi. The ‘programmed cell death’ GO term was found in up- and down-regulated genes in the resistant family at 3 hpi and 2 dpi, while it was mainly down-regulated in the susceptible family. Interestingly, the ‘natural killer cell-mediated cytotoxicity’ pathway was enriched only in the resistant fish at 2 dpi. Following infection, the host regulates the infected cells’ death to tailor the overall immune response of the surrounding environment. Necroptosis is a non-apoptotic form of cell death that has evolved to detect pathogens and promote tissue repair. It recruits immune cells but also induces inflammation, which can be detrimental to the host [46]. On the other hand, apoptotic death leads to immunologically silent responses, and its activation does not promote a significant inflammatory response, thereby preserving homeostatic integrity [47]. Apart from necroptosis, the resistant family utilizes apoptosis, programmed cell death and natural killer cell cytotoxicity in a more balanced way compared to the susceptible fish, a fact that is possibly responsible for the better control of the viral infection’s outcomes.

Viruses have direct effects on the metabolism of the cells they infect and in other host cell types. Moreover, several innate immune responses, including interferon-mediated processes, can modulate the host metabolism [48]. For example, the insulin receptor is implicated in viral responses through the modulation of T cell metabolism [49]. In the present study, the insulin-related pathways were more intensely enriched in the resistant family. The ‘insulin resistance’ pathway was only enriched at the initial time point in both families, as the ‘insulin signaling pathway’ was also up-regulated in the resistant family at 2 dpi. Interestingly, the ‘insulin secretion’ pathway was enriched at 3 hpi in the resistant family, but it was also significant at the 14 dpi time point, implying a possible role in virus carrier state establishment. Insulin signaling-related genes were also evident in *A. salmonicida*-resistant turbot [24]. Many viruses also affect metabolism by changing the nature of lipid metabolism [48], a KEGG pathway category that was mainly enriched in the NNV-resistant sea bass, but was also observed in the susceptible family. In fact, the ‘sphingolipid signaling pathway’ was enriched at all time points by the resistant family, but in the susceptible family it was found at 3 hpi and 14 dpi, while ‘sphingolipid metabolism’ was found to be enriched at 2 dpi in the resistant family and at 3 hpi in the susceptible family. The ‘glycerophospholipid metabolism was found to be enriched at 2 dpi in both families, and the ‘ether lipid’ pathway was found to be up-regulated only in the resistant family at 3 hpi. The GO term ‘lipid metabolic process’ was found to be up-regulated at 3 hpi in the resistant family and at 2 dpi in the susceptible family, while it was down-regulated at those time points in both families. The same trend in down-regulated genes was found for the ‘lipid binding’ term, which was up-regulated only at 3 hpi in the resistant family. Also, the ‘lipid droplet’ term was up-regulated at 2 dpi in the susceptible family. A modulated lipid metabolism was also observed in IPNV-infected Atlantic salmon fry [22], in the livers of *A. salmonicida*-resistant and -susceptible turbot [24], and in European sea bass re-infected with NNV [26]. It has been proven that lipogenesis is required for effective NNV infection, and the virus induces the formation of lipid droplets [50]. The present results indicate the host’s effort to down-regulate the lipid metabolic process and binding in both families, which seems more effective for the resistant family, since NNV-favorable mechanisms, like lipid droplet formation, are enhanced in the susceptible family. At the same time, however, it has been reported that the down-regulation of lipid metabolism in susceptible fish might contribute to the exacerbation of the immune response [22]; therefore, this mechanism of action needs further study. Apart from glucose and lipid metabolism, several pathogens can cause changes in vitamin and co-factors’ metabolism [48]. In our study, the GO term ‘vitamin metabolic process’ was down-regulated at the initial time point by the resistant family and at 2 dpi by the susceptible family. The susceptible family appeared to be enriched in the ‘vitamin digestion and absorption’ pathway at 2 dpi and the ‘retinol metabolism’ at 3 hpi. This trend was also observed in IPNV-susceptible salmon fry, and it was attributed to severe viremia [22]. All mentioned modulations in metabolic processes (insulin, lipids, vitamin, etc.) reflect the host’s attempt to overcome the lower energy availability due to appetite loss (as a result of the viral infection), and to restrain the production of metabolic elements that are essential for virus survival and replication at the same time. This balance seems to have a great effect on the host’s resistance to a pathogen.

The ‘oxidative phosphorylation’ pathway was found to be enriched only in the susceptible family at the 3 hpi time point, but it was enhanced in both families at 14 dpi. Interestingly, a pathway related to reactive oxygen species (ROS) was enriched in the resistant family at 3 hpi and 2 dpi, but only at the 14 dpi time point in the susceptible family. Also, the ‘antioxidant activity’ GO term was down-regulated at 2 dpi in the resistant family. Terms related to ROS production and response were also found enriched in *A. salmonicida*-resistant turbot [24]. Several RNA-related pathways were found enriched in both families, as expected, since NNV is an RNA virus. The ‘RNA polymerase’ and ‘spliceosome’ were up-regulated at all time points, except the 2 dpi time point in the susceptible family, reflecting the virus’ activity in the host cells. It is worth noting that only the resistant family utilized the ‘RNA degradation’ and the ‘mRNA surveillance pathway’ at the 2 dpi time point, in an attempt to control the RNA virus, which seems to have had a positive effect on resistance to the NNV. The same observations were found in NNV-resistant sea bass following re-infection [26].

Many transcriptome differences between resistant and susceptible sea bass were related to cytoskeleton and cell adhesion. The cytoskeleton plays a versatile role in immune processes, since its role is to support the migration of immune cells to reach the intruder, but it is also related to crucial immune receptor signaling functions that are dependent on cytoskeletal organization and dynamics. The host’s cytoskeleton is crucial for pathogen sensing, the cell-intrinsic actions of IFN-stimulated genes, phagocytes action, inflammasome activation, autophagy and cell death. It is also implicated in antigen presentation, T cell signaling and adaptive immune responses [39]. Enriched GO terms related to cytoskeleton were found in both families. ‘Cytoskeleton organization’ was found up- and down-regulated at 3 hpi and 2 dpi in the resistant family, but in the susceptible family it was down-regulated at all time points. ‘Cytoskeleton’ was up- and down-regulated at 2 and 14 dpi in the susceptible family and down-regulated in the resistant family at 2 dpi. ‘Wound healing’ was de-regulated in the resistant family at the 3 hpi time point, but it was down-regulated in the susceptible family at 3 hpi and 2 dpi. Interestingly, the regulation of the actin cytoskeleton pathway was only enriched in the resistant family (3 hpi and 2 dpi), suggesting that a more controlled action of the cytoskeleton plays a crucial role in host response to a pathogen. This finding is supported by other studies as well [24,51]. Cell adhesion plays an important role in mediating the migration of immune cells and maintaining tissue integrity, with effects on tissue healing. It is well known that several viruses use cell adhesion molecules as receptors to facilitate their entry into the cell, but the mechanisms of action are still under study [52]. The ‘cell adhesion molecules’ pathway was enriched in both families at 3 hpi and 2 dpi, while the respective GO term in the susceptible family was solely down-regulated. The ‘cell adhesion mediator activity’ term was also down-regulated in the susceptible family at 2 dpi, but it was up-regulated in the resistant fish at the 3 hpi time point. On the other hand, the ‘focal adhesion’ pathway was enriched at 3 hpi in both families and at 2 dpi only in the resistant family. A study on NNV infection in grouper larvae revealed several up-regulated genes related to cytoskeleton, adhesion molecules, and collagen synthesis, which might suppress the acute and lethal immune responses upon NNV infection [53]. Therefore, the observed cytoskeleton and adhesion profiles in sea bass may indicate a better orchestrated mechanism for overcoming infection’s effects on tissues in the resistant family, since these terms are mostly down-regulated in the susceptible fish.

Intracellular transport systems are important for the delivery of various molecules, including proteins and lipids, to the points where they are needed. The GO terms ‘vesicle-mediated transport’ and ‘cytoplasmic vesicle’ were found mostly in the resistant family’s up-regulated DEGs, at 3 hpi and 2 dpi. In contrast, the ‘vesicle-mediated transport’ was down-regulated in the susceptible family. Viruses hijack transport pathways to promote their propagation in the cell, and specifically, the betanodavirus enters host cells mainly via endocytosis involving clathrin-coated vesicles containing internalized viruses [54]. The up-regulation of the vesicle-related terms in the resistant family and the opposite trend in the susceptible fish suggest that sea bass attempt to utilize their cells normal vesicular transport pathways to defend themselves from the infection, as previously found in our NNV re-infection study of resistant and susceptible sea bass [26].

Motor proteins are accessory transport proteins that mediate the intracellular movement of cargoes (e.g., organelles, vesicles, protein complexes, etc.) along with the cytoskeleton, and viruses have developed mechanisms to hijack dyneins, kinesins and myosins cells to promote replication and propagation [55]. The ‘motor proteins’ pathway was found enriched in both families at all time points, except 2 dpi in the susceptible family. The ‘cytoskeletal motor activity’ and ‘cell motility’ GO terms were up-regulated by the resistant family at 3 hpi and 2 dpi, were also found in down-regulated DEGs of the resistant family. In the susceptible family, these terms were only found to be down-regulated at 3 hpi, and at all time points, respectively. Our findings are in partial agreement with those derived from the transcriptome analysis of *A. salmonicida*-resistant turbot [24]. Moreover, the ‘muscle system process’ was enriched in both up- and down-regulated DEGs of resistant sea bass (3 hpi and 2 dpi), but again, in the susceptible family, it was only down-regulated (3 hpi and 14 dpi). This profile seems consistent with abnormal swimming, which is one of the disease symptoms and was more evident in the susceptible family, as reported before [22,26].

Finally, generic viral disease (‘Influenza A’ and ‘Coronavirus disease—COVID-19’) pathways were enriched in both families at all time points, in accordance with similar studies [22,26].

A total of five hub genes that are directly related to host immune responses in both the NNV-resistant and the NNV-susceptible sea bass families were identified by the PPI analysis, and are possibly related to NNV resistance in European sea bass. The only immune-related hub gene at the 3 hpi time point was the CD59 glycoprotein (*cd59*). CD59 is a small widely distributed glycoprotein that belongs to the leukocyte antigen 6 (Ly-6) family and binds to cell membrane phospholipids. It has a regulatory effect on the complement system, being involved in complement inhibition and the formation of the cytolytic membrane attack complex (MAC), and acts as a signaling molecule that can participate in T cell activation and insulin secretion [56]. Teleostean CD59s have been identified in zebrafish, Nile tilapia, large yellow croaker, tongue sole and orange-spotted grouper [57]. It is interesting that several viruses (e.g., human immunodeficiency virus (HIV), human cytomegalovirus (HCMV), and others) incorporate in their viral envelope host cellular factors, including CD59, which normally protects cells against complement lysis [43]. At the 2 dpi time point, the following hub genes were found down-regulated in the resistant family: interleukin-1 beta (*IL-1b*), dual specificity protein phosphatase 1 (*dusp1*) and nuclear receptor subfamily 4 group A member 1 (*nr4a1*). IL-1b is an important mediator of the inflammatory response, and is produced by a wide range of cell types early after the activation of host pattern recognition receptors (PRRs), enabling a response to infection by the induction of a reactions cascade that leads to inflammation. IL-1b is involved in phagocyte migration, macrophage activity and lymphocyte activation, and it induces immune-suppressive cytokines such as IL-10 and prostaglandins [58]. Several studies have proven that IL-1b plays a crucial role in the host immune response to NNV infection [34,36]. The dual-specificity phosphatase (DUSP) family is involved in the maintenance of immune cell homeostasis, inflammatory responses, apoptosis and metabolic regulation [59]. Dusp1 has emerged as a central mediator in the resolution of inflammation; it is implicated in T-cell activation, and regulates innate immunity responses to different stimulations, including pathogen infections. In fish, dusp1 has been found deregulated in Japanese flounder following polyI:C and LPS stimulation, or *E. tarda* infection, in carp infected with Cyprinid herpesvirus 2, and in *Epinephelus coioides* infected with Singapore grouper iridovirus (SGIV) [60]. The nuclear receptor subfamily 4A (*Nr4a*) receptor family members are linked to various physiological and pathological processes, including lymphocyte development, metabolism, the promotion of immunological tolerance, cell-cycle regulation, inflammation, and apoptosis [61]. The role of nr4a1 (or nur77), which has been studied in model organisms, is currently under investigation in fish [61,62]. At the same time point, one hub gene was found down-regulated in the susceptible family i.e., the alpha-2-HS-glycoprotein (*ahsg1*). Ahsg (also termed fetuin-A) belongs to the ‘acute phase proteins’ (APPs), and its expression is negatively regulated by several pro-inflammatory cytokines such as TNF, IL-1, IL-6 and IFN-γ during infection or other inflammatory illnesses. Moreover, fetuin-A is involved in biological activities such as bone and calcium metabolism and insulin signaling pathway regulation, acting as a protease inhibitor, inflammatory mediator, anti-inflammatory partner, and atherogenic and adipogenic factor [62,63].

Two complement hub genes were down-regulated in the susceptible family at 2 dpi—complement component C5 (*c5*) and complement component C8 alpha chain (*c8a*). C5 is a part of the innate immune system and plays an important role in virus neutralization, inflammation, host homeostasis and host defense against pathogens [43]. It is involved in NNV pathogenesis and was also found down-regulated in NNV-infected orange-spotted grouper [53]. C8a is involved in the formation of the membrane attack complex (MAC) and effects cell membrane integrity. Even though it is poorly studied, it was found that C8 proteins in zebrafish might function mainly in the neuroendocrine system [56], an interesting clue, since NNV is a neurotropic virus. It is noteworthy that CD59 plays a role in binding to C8a in mammals, and it has been proposed that they also interact in fish [56]. As was mentioned, CD59 is heavily down-regulated at the 3 hpi infection time point in the resistant family; therefore CD59–C8a interaction may play an important role in the NNV infection of sea bass, and needs further study.

Apart from coagulation cascade enrichment in the susceptible family, eight hub genes related to the coagulation system were identified in both families, at different time points. At the 3 hpi time point, the resistant family had two deregulated genes—antithrombin III (*serpina1*) and multiple coagulation factor deficiency protein 2 (*mcfd2*). Antithrombin is a natural anticoagulant that interacts with activated proteases of the coagulation system and heparan sulfate proteoglycans on the surfaces of cells. It is implicated in anti-inflammatory signaling responses, immunological processes and tissue damage, and has direct antimicrobial effects. It fights viruses either by the induction of anti-viral signaling in infected cells or by inhibiting the virus’ entry [64]. In fish, a recent study reported that SERPINA1 can inhibit Grass carp reovirus (GCRV) infection through interaction with the coagulation factor CF2 [65]. MCFD2 is a small, soluble protein that forms a Ca^2+^-dependent complex with LMAN1 and transports coagulation factors V and VIII from the endoplasmic reticulum (ER) to the ER–Golgi intermediate compartment (ERGIC) [66]. At the following time point (2 dpi), the resistant family appeared to have one hub gene related to blood (heme oxygenase-like; *hmox1a*) and the susceptible family had six hub genes involved in coagulation and blood disorders (plasminogen (*plg*), uricase (*uox*), heparin cofactor II (*serpind1*), hyaluronan-binding protein (*habp2*), alpha-2-antiplasmin (*serpinf2a*)). Heme oxygenase catalyzes heme degradation to biliverdin, iron and carbon monoxide. Its expression is induced in response to reactive oxygen species and inflammatory mediators, with antioxidant, anti-inflammatory and immunosuppressive consequences for the host [67]. In fish, *hmox1a* was found up-regulated in salmon infected with IPNV [68]. The plasminogen (PLG) activation system is composed of a series of serine proteases, inhibitors, and several binding proteins, which control the temporal and spatial generation of the active serine protease plasmin. The plasmin/plasminogen system is an essential part of the innate immune responses, mediating inflammatory processes. Plasminogen has been found down-regulated in *Piscirickettsia salmonis*-resistant Atlantic salmon, and the authors suggest that its lower expression elicits a constrained inflammatory response; therefore, down-regulation by the susceptible family at the 2 dpi time point may reflect the fish’s attempt to better control the viral infection [69]. Alpha-2-antiplasmin is a plasmin inhibitor, and has various functions including cell differentiation, cytokine production, immune system regulation, endothelial homeostasis, and extracellular matrix metabolism. In fish, it has been found activated following vaccination, indicating the host’s attempts to avoid excessive localized and systemic plasmin generation [70]. Heparin cofactor (HCII) is a single-chain glycoproteinous serpin that resembles antithrombin III, and as a serine protease, it is involved in important biological processes such as digestion, blood clotting, immune activation and cell differentiation. A recent study on rock bream challenged with various immune stimulants found that it was down-regulated, indicating an inhibitory role against immune- and coagulation-related proteins [71]. Uricase is the primary enzyme that converts uric acid (which is the final product of purine metabolism) to allantoin. In humans, urate crystals are considered a critical natural endogenous immune response trigger, and are implicated in inflammation [72]. Hyaluronan binding protein 2 or factor VII-activating protease is involved in fibrinolysis, inflammation and tissue remodeling via the regulation of the uPA/plasminogen system in both mammals and fish [73].

Four hub genes related to metabolism were found strongly down-regulated by the susceptible family via PPI analysis at the 2 dpi and 14 hpi time points. More specifically, two alanine-glyoxylate and serine-pyruvate aminotransferases (*agxt*) were down-regulated at 2 dpi, i.e., agxta and agxtb. Agxt genes encode an enzyme called alanine-glyoxylate and serine-pyruvate aminotransferase, which is mainly found in liver cells, within peroxisomes. Serine-pyruvate aminotransferase converts glyoxylate to glycine and is involved in glyoxylate detoxification. Both agxta and agxtb have been reported following genomic, transcriptomic and proteomic analyses in fish focused on host–pathogen interactions, developmental procedures and stress responses, but their specific functions remain unclear [74]. At the 14 dpi time point, the two down-regulated hub genes were glyceraldehyde-3-phosphate dehydrogenase (*gapdh*) and aldolase a, fructose-bisphosphate, b (*aldob*). Glyceraldehyde-3-phosphate dehydrogenase (gapdh) plays multiple roles in energy metabolism as well as in transcription, apoptosis and neurodegenerative disorders, both in mammals and in teleost fish [75]. Fructose 1,6-bisphosphate aldolase can hydrolyse fructose 1,6-bisphosphate to become gapdh and dihydroxyacetone phosphate, and it has functions associated with virulence, such as adhesion to host cells, plasminogen binding, tissue invasion, and cell invasion. It has been tested as a broad-spectrum vaccine against pathogenic bacteria in aquaculture, but has not been associated with viral infections in fish [76].

Three hub genes involved in genetic information processing were found in the resistant family at 3 hpi and 2 dpi. At 3 hpi, the pre-mRNA 3′-end-processing protein WDR33-like (*wdr3*) was up-regulated. WDR33 is a subunit of the cleavage and polyadenylation specificity factor (CPSF), which is responsible for the endonucleolytic cleavage process followed by polyadenylation in eukaryotic pre-mRNAs. Viruses selectively disrupt host 3’-end processing, while viral transcription and 3’-end processing remain intact, affecting the antiviral response [77]. In fish, the role of WDR33 remains unclear. Two more hub genes corresponding to transcription factors were down-regulated at 2 dpi. The heart- and neural crest derivatives-expressed protein 2 (*hand2*) is a transcriptional regulator that interacts with the regulatory regions of many genes that function during differentiation, including neuronal differentiation. It has been found to interfere with antiviral drugs against Borna disease virus, a neurotropic virus [78]. Paired like homeobox 2Bb (*phox2bb*) is a transcription factor expressed in enteric progenitors and differentiating neurons during enteric nervous system development [79].

Six hub genes related to transport were up-regulated in the resistant family at the initial time point: Golgi SNAP receptor complex member 2 isoform X2 (*gosr2*), SCFD1 protein (*scfd1*), protein transport protein Sec23A-like (*sec23a*), vesicle-trafficking protein SEC22c (*sec22c*), protein cornichon homolog 1 (*cnih1*) and vesicular integral-membrane protein VIP36 (*lman2*). The Golgi SNAP receptor complex member 2 protein (gosr2/membrin) is involved in intracellular vesicles trafficking from the endoplasmic reticulum (ER) to Golgi and has been associated with a spectrum of neurologic conditions [80]. Sec-1 family domain-containing 1 protein (scfd1) encodes a member of the Sec1/Munc18 family of proteins, which cooperate with soluble N-ethylmaleimide-sensitive factor attachment protein receptor (SNARE) complexes in membrane fusion events in the secretory and endo-lysosomal pathways, as well as ER to Golgi ante- and retrograde transport [81]. Interestingly, SCFD1 is considered an amyotrophic lateral sclerosis risk factor, and its down-regulation is related to motor dysfunction in Drosophila [82]. Sec22 is also an important SNARE protein involved in membrane fusion in eukaryotes. It is localized to ER and Golgi and helps in the anterograde and retrograde transport of vesicles, playing a significant role in the secretion of cytosolic proteins by secretory autophagy and participating in autophagosome–vacuole fusion [83]. The protein transport protein Sec23 is one of the components in the coat protein complex II, and regulates the transportation of proteins and lipids from ER to the Golgi apparatus in cells. SEC23A has effects on ER stress, and is a master regulator of budding and fusion, helping to facilitate the formation of the autophagosome [84]. The cornichon homolog 1 protein (cnih1) belongs to cornichon family proteins, which are ER membrane proteins and act as cargo receptors for vesicles’ transport to their target membrane. Recently, members of the CNIH and SEC23 protein families have been found to interact in moss [85]. The vesicular integral membrane protein of 36 kDa (VIP36; lman2) is an L-type lectin. It functions as a quality control cargo receptor, identifies and translocates glycoproteins, has a regulatory role in phagocytic activity both on the cell surfaces and in extracellular settings, and has been associated with innate immunity in crustaceans [86].

Fifteen hub genes related to motility functions were identified in both families. In the resistant family, three kinesin-related genes were found deregulated at 2 dpi—kinesin-like protein KIF1A isoform X1 (*kif1aa*), kinesin family member 5Aa isoform X1 (*kif5aa*) and kinesin-like protein KIF20A isoform X1 (*kif20a*). Kinesins are molecular motor proteins that hydrolyze adenosine triphosphate (ATP) to move along, or actively remodel, microtubules within cells. KIF1A is a critical cargo transport motor within neurons, and it is associated with developmental and degenerative neurological diseases [87]. Recent studies on pseudorabies virus (PRV), which is a neuroinvasive pathogen, have shown that PRV infection disrupts the synthesis of KIF1A and simultaneously promotes the degradation of intrinsically unstable KIF1A proteins by proteasomes, because the ectopic expression of KIF1A proteins reduces the number of PRV particles transported in the anterograde direction [88]. Kinesin family member 5A also mediates the transport of various cargoes, such as organelles, synaptic vesicle precursors, neurotransmitter receptors, and mRNAs, across neurons, playing important roles in neuronal functions. In fish, KIF5A has also been found to be involved in immune response following *Streptococcus parauberis* infection [89]. KIF20A regulates cell division via central spindle assembly and cleavage furrow formation. Studies on influenza A virus (IAV) suggest that KIF20A is highly involved in the viral replication process, since it increased viral protein levels in IAV-infected cells by regulating the initial entry stage during viral infection, and the use of a KIF20A inhibitor significantly suppressed viral replication in mice [90].

In the susceptible family, two myosin-related genes, four troponin-related genes and two actin/actinin genes were found significantly down-regulated at 3 hpi—myosin-7 myosin heavy chain 7 (*myh7*), myosin, light polypeptide 3, skeletal muscle (*mylz3*), troponin T, cardiac muscle isoforms isoform X10 (*tnnt2a*), troponin I, fast skeletal muscle-like (*tnni2a.4*), troponin T, fast skeletal muscle isoforms-like (*tnnt3a*), troponin I (*tnni1a*), alpha-actinin-3 (*actn3a*), actin, and alpha cardiac (*actc1a*). Τitin-like isoform X1 (*ttn.2*) and sarcoplasmic/endoplasmic reticulum calcium ATPase 1 (*atp2a1*) were also strongly down-regulated at 3 hpi. Myosins are actin-associated motor proteins that are involved in a variety of cellular activities, including signal transduction, intracellular transport and cell division, and it has been proven that they play significant roles in viral infections. Myosin light chain 3 (MYL3) is found in muscle tissues (heart and skeletal muscles), and is fundamental in muscle contraction, forming the myosin complex with myosin heavy chain and actin filament sliding. Recently, it has been suggested that myosin light chain 3 serves as a receptor for nervous necrosis virus entry into host cells via the macropinocytosis pathway [91]. Myosin heavy chains (MYHs) participate in organelle movement and mitosis, extracellular phagocytosis, motility, fertilization, absorption, intracellular signal transmission and changes in cell morphology. In fish, myh7 has been proposed as a candidate molecular marker for enhanced growth in selective breeding [92]. Alpha-actinins are structural proteins that act as glue proteins, and are responsible for the integrity and maintenance of the sarcomere by linking titin and actin filaments. In fish, the alpha-actinin-3 protein has been found down-regulated in response to oxygen stress [93]. The cardiac α-actin (ACTC1) forms a thin contractile filament with tropomyosin and troponins. Troponins are an important component of the thin filaments that allow muscles to contract in response to calcium. The troponin complex is composed of troponin C (TnC), troponin I (TnI), and troponin T (TnT). TnT interacts with TnC, TnI, tropomyosin and actin, and has been found to play a role in invertebrates’ resistance to pathogens through apoptosis [94]. Troponin I is the actomyosin ATPase inhibitory subunit of the troponin complex, and even though it is adequately characterized in mammals, its functions in fish remain unclear. Titin is the largest sarcomeric protein expressed in muscles, and plays a crucial structural and functional role in sarcomeres. Zebrafish (*Danio rerio*) and Medaka (*Oryzias latipes*) titin has great structural similarity to human titin [95]. Sarco/endoplasmic reticulum calcium-ATPase (SERCA) is located in the endoplasmic reticulum and transports Ca^2+^ from cytosol to intracellular compartments. It also shows signal transduction, apoptosis, exocytosis, cell motility and transcription functions. Paramyxovirus replication was found to be facilitated by SERCA; therefore, it seems to play a critical role in virus replication control [96]. Finally, four hub genes were found down-regulated in the susceptible family at 14 dpi, with three of them belonging to actin and actinin families, which were also deregulated at 3 hpi, such as alpha cardiac muscle actin 1 (*actc1b*). Alpha-actinin-2 acts (*actn2b*) is also similar to alpha-actinin-3, which was found downregulated at 3 hpi, but studies on fish and the infection of both actinins have mostly focused on zebrafish and myopathies [97]. Beta-actin plays a key role in maintaining cytoskeletal structure, cell motility, cytokinesis, endocytosis, and cell adhesion, and has been widely used as a “housekeeping” gene. However, there are reports that its expression is altered in response to several stimulations, including HSV-1 infection [98]. The ryanodine receptor (RyR1) intracellular Ca^2+^ channel plays a central role in excitation–contraction coupling by releasing Ca^2+^ from the sarcoplasmic reticulum. It is mainly expressed in skeletal muscle, but it is also found in neurons, smooth muscle cells and immune cells, specifically in B-lymphocytes and dendritic cells. In fish, RyRs are involved in non-dioxin-like ortho-substituted polychlorinated biphenyl (NDL PCBs), which are potential neurotoxic compounds [99]. All mentioned hub genes could represent interesting markers for disease resistance, as well as markers for disease progression, since they are different for each family at each time point. However, their potential use as markers needs to be further studied in more sea bass tissues, and to be validated in natural infections as well.

Overall, the NNV-resistant fish seem to control their response to viral infection more efficiently compared to the susceptible fish. The resistant fish display higher levels of interferon-related pathway elements, cytokines, antigen presentation and T-cell activity, combined with a controlled inflammatory response and more active proteasome and lysosome functions. Resistant fish also utilize more apoptosis and programmed cell death, while the susceptible fish mainly rely on necroptosis. On the other hand, the susceptible fish appeared to show high immune responses at the early infection stages, but also showed the high expression of inflammatory pathway members and the enrichment of complement and coagulation pathways. Insulin metabolism is regulated better in the resistant family, and the control of lipid metabolic processes seems less effective in susceptible fish. Cytoskeleton organization- and cell adhesion-related pathways were mostly down-regulated in the susceptible fish, thus their role in virus infection restriction is crucial. In relation to that, intracellular transport and motor proteins are utilized more efficiently by resistant fish. All mentioned observations suggest that the mechanism that offers pathogen resistance to a host has many aspects, which can be exploited in order to develop more efficient approaches to fight diseases in aquaculture.

## 4. Materials and Methods

### 4.1. Ethics Statement

Experimentation was performed in the Laboratory of Ichthyology-Aquaculture and Aquatic Animal Health (ICHTHYAI) (Government Issue 1255/28-4-2016). ICHTHYAI has been granted all the required permits for producing (EL 83 BIObr 01), supplying (EL 83 BIOsup 01) and experimenting on aquatic organisms (EL83 BioExp 01), according to the Presidential Decree 56/2013 confirming to Directive 2010/63/ΕΕ (Decision No 2700/11-3-2022 of the competent Regional Veterinary Authority). Fish were euthanized using a procedure listed on the appropriate license and the protocol for the experimental infection performed in this study was approved by Decision No 5690/11-05-2022 of the competent Regional Veterinary Authority.

### 4.2. Experimental Fish and Nervous Necrosis Virus

Healthy, non-vaccinated fish were used throughout the experimentation. Two families of European seabass with different levels of resistance to NNV (i.e., one NNV-resistant family (R) and one NNV-susceptible family (S)), as previously determined on a family-based breeding program [3], were chosen. The present study was performed in the context of a larger project, parts of which have been previously reported [26,36]. Briefly, fish were transported to ICHTHYAI by Nireus Aquaculture S.A. A total of 300 randomly selected fish from both families (weight: 125.85 ± 29.8 g) were acclimatized for 3–4 days in 1 m^3^ cylindroconical fiberglass tanks connected to a closed recirculated sea water system, with 20 m^3^ total volume capacity. Fish were reared in a 12 h light:12 h dark photoperiod and were fed with 1–2% of their biomass of commercial (Feedus, Blueline) diet 3 times a day with 6 h intervals. Water was recirculated via a 14 m^3^·h^−1^ sea water pump, filtered via a sand filter, disinfected via 5 × 39 W UV-C lamps, treated in a biological filter and aerated via air stones connected to five 150 L·h^−1^ air pumps. The seawater temperature was maintained at 22–23 °C during the acclimatization period. Salinity was 3.8–3.9‰, dissolved O_2_ was maintained above 4.8 mg·L^−1^, total ammonia nitrogen and nitrite were kept below 0.05 ppm and 0.5 ppm, respectively, and nitrate levels were below 40 mg·L^−1^. pH ranged between 7.9 and 8.1. Temperature, dissolved oxygen and nitrogen metabolites were measured daily, while salinity and pH were measured on a weekly basis. Each fish carried an individual electronic tag, which was previously inserted into their abdominal cavity.

Nervous necrosis virus (NNV) was originally isolated from naturally infected *D. labrax* (genotype: RGNNV [100]) and NNV propagation was performed as previously described [101]. Briefly, fish brains were homogenized in EMEM (Eagle Minimum Essential Medium; Sigma-Aldrich, Steinheim, Germany) or Leibovitz L15-medium (Biochrom, Berlin, Germany) containing 10% FBS (Fetal bovine serum; Biochrom). The homogenates (10% *w*/*v*) were centrifuged (4000× *g*, 15 min, 4 °C) and the resulting supernatant was passed through 0.22 μm filters (Whatman PTFE, 0.22 μm; GE healthcare, Buckinghamshire, England) before inoculation on cell cultures. Following inoculation, the SSN-1 cells (The European Collection of Animal Cell Cultures, Salisbury, UK) were grown at 26 °C in Falcon Primaria cell culture flasks (Becton Dickinson Lab-ware, Franklin Lakes, NJ, USA) containing Leibovitz’s L15-medium, supplemented with 10% FBS, 100 u·mL^−1^ penicillin, 100 μg·mL^−1^ streptomycin and 2 mM glutamine (Gibco, Paisley, UK). Virus titration was performed via the end-point titration method. Serial dilutions of viral suspensions in EMEM–10% FBS were added in a 96-well plate seeded with monolayers of SSN-1 cells and incubated at 26 °C for 6 days. During this period, the cell monolayers were observed for the appearance of a cytopathic effect (CPE) and the final titer, expressed as TCID_50_·mL^−1^, was estimated [102].

### 4.3. Nervous Necrosis Virus Challenge

A summary of the experimental setup is presented in Figure 1. On day 7 after the initial rearing, fish were stocked in an allocated tank system (in triplicate per family/per condition) in groups of 20 and 50 for family R and family S, respectively. The temperature was gradually raised to 27.0–27.2 °C and remained constant throughout the experimentation. Before the infection experiment, total RNA was extracted from three randomly selected individuals’ brains and amplified with an NNV RT-qPCR assay using the Quantitect Probe RT-PCR master mix (Qiagen; Hilden, Germany) to ensure that the specimens were not infected. Sea bass were challenged by intramuscular injection in the dorsal muscle with 7 × 10^6^ TCID_50_·mL^−1^ of nodavirus-containing supernatant (200 μL; serially diluted with PBS from a 1 × 10^11^ TCID_50_·mL^−1^ nodavirus-containing supernatant stock). As the negative control group, uninfected sea bass from the same families were mock-challenged with 200 µL PBS. Fish were monitored twice a day and mortalities were recorded up to 28 days’ post infection (dpi). At three specific time points, i.e., 3 h post infection (hpi), 2 dpi and 14 dpi, 9 fish (3 samples from each tank) were randomly selected. These time points were rationally chosen as times points with higher immune responses in European sea bass experimentally challenged with NNV based on selected immune-related genes’ expression levels [36]. Before sampling, the specimens were anesthetized with 0.2% phenoxyethanol and weighed. Following this, head kidney and brain tissues were removed aseptically and were either subjected to RNA isolation immediately or stored in RNAlater (Qiagen) at −80 °C.

### 4.4. RNA Extraction

The total RNA from the brain and head kidney was extracted using a Trizol-based protocol to perform total RNA sequencing. Tissues (~75 mg) from NNV-infected and non-challenged fish at each time point were homogenized using the TissueLyzer mechanical homogenizer with 5 mm steel beads (Qiagen). The TRIzol^®^ Reagent solution (Invitrogen, Carlsbad, CA, USA) was used for total RNA extraction, following the manufacturer’s instructions. RNA quantity and quality were assessed by spectrophotometry (NanoDrop 2000) using a Qubit 2.0 fluorometer (Thermo Fisher Scientific, Waltham, MA, USA) and the RNA integrity number (RIN) was measured using a 2100 Bioanalyzer instrument (Agilent Technologies, Santa Clara, CA, USA). Only RNA samples of high quality (RIN ≥ 6.5) were used for constructing the cDNA libraries, and each sample’s transcriptome was analyzed individually.

### 4.5. NNV Viral Load Testing

Randomly selected fish from each tank were tested for the presence of NNV at the specified time points. A generic one-step RT-qPCR assay was applied for the determination of viral load in total RNA from fish brain tissue, as described before [103].

### 4.6. cDNA Library Construction, Sequencing and Transcriptome Mapping

Τhe sequencing libraries were constructed using the Ion Total RNA-Seq Kit v2 kit (Thermo Fisher Scientific). The analysis comprised 45 individual samples; 2 genotypes (resistant, susceptible) × 2 challenge states (challenged, control) × 3 time-points (3 hpi, 2 dpi, 14 dpi) × 3–5 biological replicates. The 45 libraries were prepared according to the manufacturer’s instructions and then sequenced using Ion Torrent technology (Ion S5XL platform, Thermo Fisher Scientific). Raw data were processed to remove adapters by Trimmomatic, and were normalized for inherent systematic or experimental biases, using the Bioconductor package DESeq2 (version 3.12). The reads were aligned to *Dicentrarchus labrax* reference genome (sea-bass_V1.0-GCA_000689215.1) [30] with hisat2 and bowtie2. Post-mapping quality control was assessed with the Bioconductor package metaseqR2 (version 3.12) [104]. All raw data reads are available in the NCBI database (BioProject accession no. PRJNA1030357).

### 4.7. Differential Expression and Functional Enrichment Analysis

Basic differential expression analysis was performed with the Bioconductor packages metaseqR2 and/or edgeR (version 3.12) [104,105]. A transcript was considered as differentially expressed if the adjusted *p*-value (FDR) threshold was less than 0.05 (significance level) and the log2-transformed fold change (log2FC) was more than |2|. The function of the identified differentially expressed transcripts was analyzed in OmicsBox software (Version 3.2.4) by first using BLASTX against an NCBI non-redundant (NR) database to search for the possible top hit proteins (accessed on 11 July 2024). To obtain high-quality results, the ‘*Actinopterygii*’ (Taxid:7898) taxonomy filter was applied. Thereafter, blasted sequences were subjected to gene ontology (GO) mapping and annotation with default parameters. GO functional enrichment and pathway analysis were carried out by BLAST2GO (BioBam Bioinformatics, Valencia, Spain) using the total transcripts dataset as the reference background. The annotated DEGs were subjected to Fisher’s exact test and were considered significantly enriched in GO terms when their Bonferroni adjusted *p*-value was less than 0.05. The results were reduced to the most specific terms. Enriched KEGG pathways were determined by Gene Set Enrichment Analysis (GSEA), with adjusted *p*-value less than 0.05. All bar plots and pathway summaries were generated using the SRplot platform [106].

### 4.8. Construction and Analysis of PPI Networks and Functional Annotation

To further investigate the relationships between resistance-related genes in both families, the Search Tool for the Retrieval of Interacting Genes/Proteins database (STRING v12.0; http://www.string-db.org, accessed on 24 April 2025) was used to construct their PPI network. All blasted differentially expressed transcripts at each time point were manually curated and translated to reference gene names orthologous to the model organism, zebrafish (*Danio rerio*; Taxid: 7955). The protein–protein interaction analysis was performed with a confidence interaction score of 0.4, followed by k-clustering analysis using the default parameters. Cytoscape (3.10.3, accessed on 24 April 2025) was used to analyze the PPI network with the CytoHubba plugin (https://apps.cytoscape.org/apps/cytohubba, accessed on 24 April 2025) in order to identify hub genes based on the maximal clique centrality (MCC) algorithm. MCC was reported to be the most effective algorithm in identifying hub genes with increased sensitivity and specificity [107]. In the present study, the genes with the top 10 MCC scores were considered as hub genes.

### 4.9. Real-Time Quantitative Polymerase Chain Reaction (qPCR) Validation

Real-time PCR assays were carried out to confirm the differential expression data of transcriptome analysis. The specific primers used for the expression analysis of genes from both families (Appendix A) were designed utilizing Primer-Blast (https://www.ncbi.nlm.nih.gov/tools/primer-blast/, accessed on 7 July 2024), IDT_Primer Quest (https://www.idtdna.com/pages/tools/primerquest, accessed on 7 July 2024), and Primer3Plus (https://www.primer3plus.com/, accessed on 7 July 2024). All qRT-PCR primers were designed according to the guidelines regarding the minimum information needed for the publication of the qRT-PCR experiment (MIQE). The beta-actin gene was chosen in our study as the reference gene. Total RNA was used as a template to synthesize cDNA using the QuantiNova Reverse Transcription kit (Qiagen) according to the manufacturer’s instructions. Approximately 5 μg of RNA was used as the input material. Real-time PCR reactions were performed with QuantiNova SYBR Green PCR (Qiagen) using 1 μL of a 1:10 dilution of cDNA. Primers for all genes were used at 500 nM. The thermal conditions used were as follows: 2 min at 95 °C of pre-incubation followed by 40 cycles at 95 °C for 10 s and 60 °C for 30 s. An additional temperature ramping step was utilized to produce melting curves from 62 to 95 °C to verify the amplification of a unique single product on all samples. All reactions were performed in technical triplicates using a RotorGene Q PCR Detection System (Qiagen). The quantification was performed according to the comparative C_T_ method [108]. The value for each experimental condition was expressed as normalized relative expression, calculated in relation to the values of the control group and normalized against those of the reference gene (by its geometric average).

## 5. Conclusions

The present study focused on the transcriptome analysis of two European sea bass families having different levels of resistance to nervous necrosis virus, an aquaculture-threatening viral pathogen. RNA-seq analysis was performed in the sea bass head kidney at three time points, representative of the early stage of infection (3 hpi), the 2 dpi time point at which the host immune response has reached a more stable level of gene expression, and the 14 dpi time point at which the host seems to act as a virus carrier. The analysis revealed a wide range of DEGs in both families, varying at each time point. Gene ontology enrichment, pathway analysis and protein–protein interaction analysis suggest that the resistance mechanism involves immune responses, inflammation, complement and coagulation cascades, metabolic pathways, and cytoskeleton and motor protein functions, which are regulated more efficiently by the resistant family. Our future plans are focused on the further integration of the present findings with other organ transcriptome analyses (e.g., brain, spleen, liver) based on RNA-seq data that are or will be publicly available, in order to find pathways implicated in sea bass resistance (from a ‘universal’ point of view). For example, from our and other studies’ results, it seems that not only immune-related but also motor protein- or lipid metabolism-related pathways are essential for conferring resistance to NNV. By combining these results with those of genomic and/or proteomic analyses, we expect to find more specific key gene(s) and be able to apply this integrated knowledge to selective breeding. In conclusion, the present work provides insights into the ways that NNV-resistant and -susceptible European sea bass reacts to the threat of viral infection through time. Further analyses of the obtained data could elucidate the specific molecular pathways that control disease resistance, and genetic selective breeding strategies could be improved by the development of specific genes or proteins as biomarkers.

## Figures and Tables

**Figure 1 ijms-26-09220-f001:**
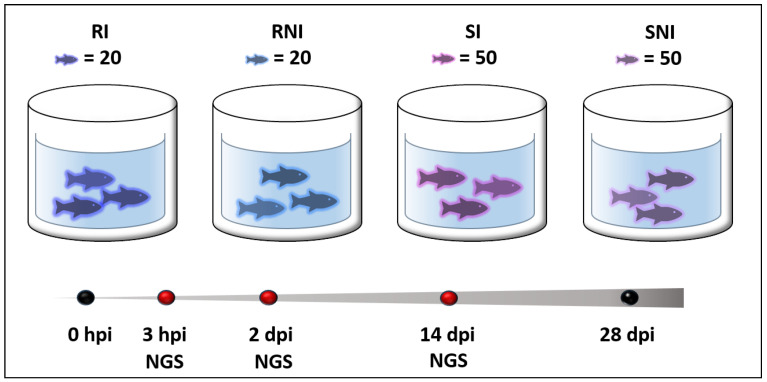
Experimental setup (RI, resistant infected fish; RNI, resistant non-infected fish; SI, susceptible infected fish; SNI, susceptible non-infected fish; hpi, hours post infection; dpi, days post infection; NGS, next-generation sequencing).

**Figure 2 ijms-26-09220-f002:**
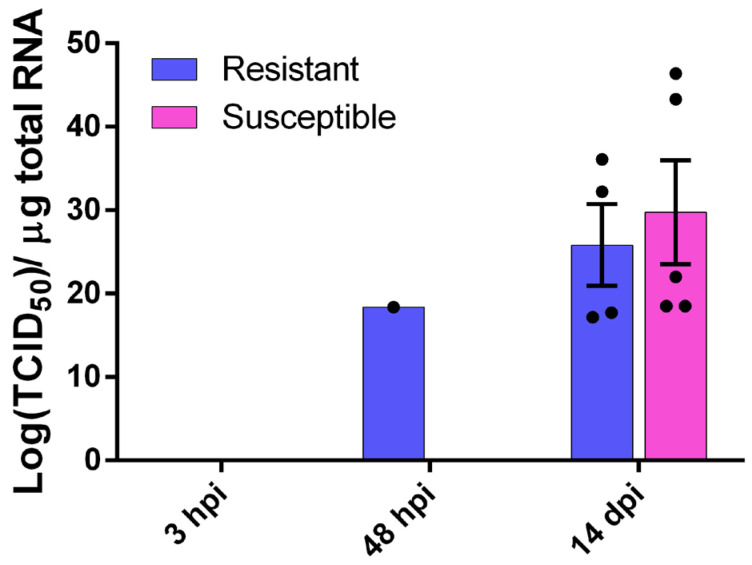
Absolute RNA1 quantification in brains from resistant and susceptible European sea bass challenged with NNV. Results are mean ± SEM (*n* = 5).

**Figure 3 ijms-26-09220-f003:**
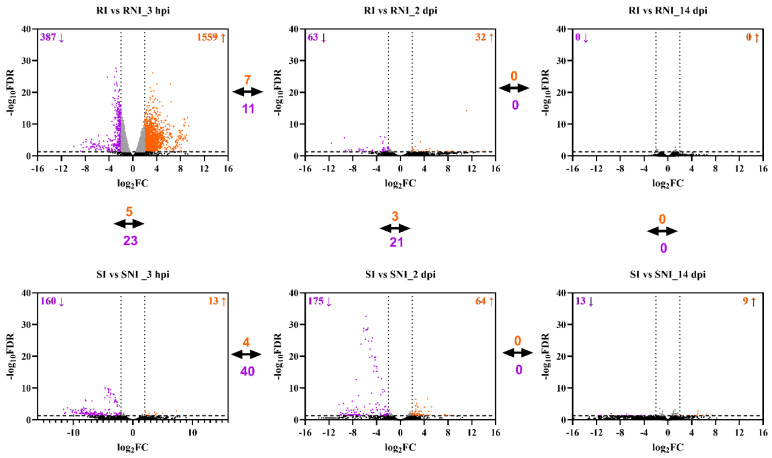
Volcano plots of gene expression response in resistant and susceptible families following NNV challenge. Volcano plots of the log_2_ fold change (FC) vs. log_10_ false discovery rate (FDR) of every transcript for each of the six comparisons are shown. The significantly up-regulated genes in each comparison (log_2_FC > 2, FDR > 0.05) are shown in orange, while the significantly down-regulated genes in each comparison (log_2_FC < −2, FDR > 0.05) are shown in purple. The black dots depict all genes with FDR < 0.05 and the grey dots represent all genes with −2 < log_2_FC < 2 and FDR > 0.05. The dashed lines correspond to log_2_FC = −2 or 2, and FDR = 0.05. The numbers of significant up-regulated (orange) and down-regulated (purple) transcripts for each comparison are shown in the corners of each volcano plot. The numbers of common significant down-regulated and up-regulated genes between the two families at each time point are shown in the in the middle of the figure, i.e., below the RI vs. RNI and above the SI vs. SNI volcano plots (NNV, nervous necrosis virus; RI, resistant infected fish; RNI, resistant non-infected fish; SI, susceptible infected fish; SNI, susceptible non-infected fish; hpi, hours post infection; dpi, days post infection).

**Figure 4 ijms-26-09220-f004:**
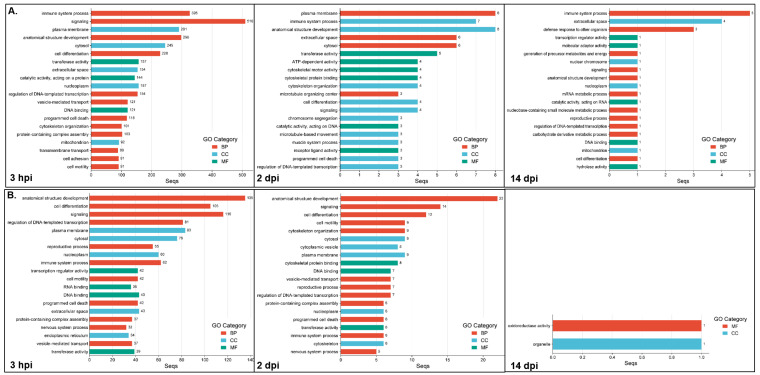
The top 20 gene ontology (GO)-enriched terms in the RI vs. RNI groups: (**A**) Up-regulated and (**B**) down-regulated DEGs. BP, biological process; MF, molecular function; CC, cellular component; RI, resistant infected fish; RNI, resistant non-infected fish; hpi, hours post infection; dpi, days post infection.

**Figure 5 ijms-26-09220-f005:**
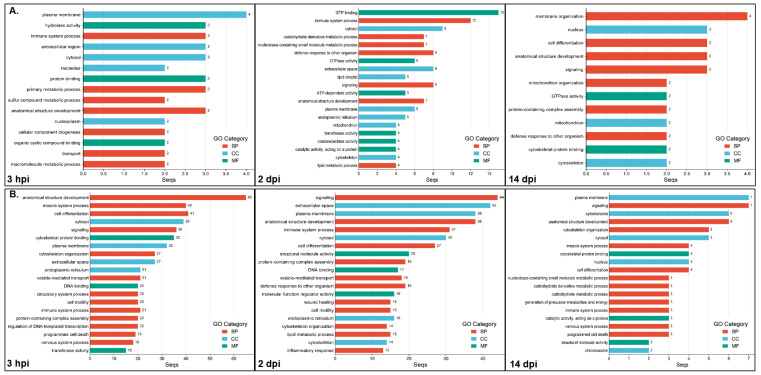
The top 20 gene ontology (GO)-enriched terms in SI vs. SNI groups: (**A**) Up-regulated and (**B**) down-regulated DEGs. BP, biological process; MF, molecular function; CC, cellular component; SI, susceptible infected fish; SNI, susceptible non-infected fish; hpi, hours post infection; dpi, days post infection.

**Figure 6 ijms-26-09220-f006:**
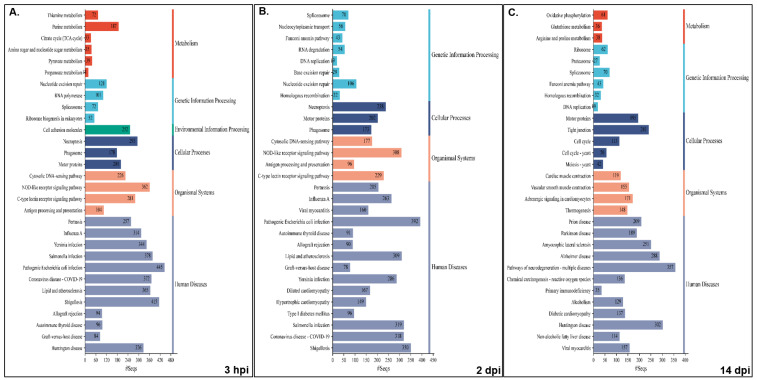
The top 30 enriched KEGG pathways in the RI vs. RNI groups. (**A**) 3 hpi, (**B**) 2 dpi, and (**C**) 14 dpi. RI, resistant infected fish; RNI, resistant non-infected fish; hpi, hours post infection; dpi, days post infection.

**Figure 7 ijms-26-09220-f007:**
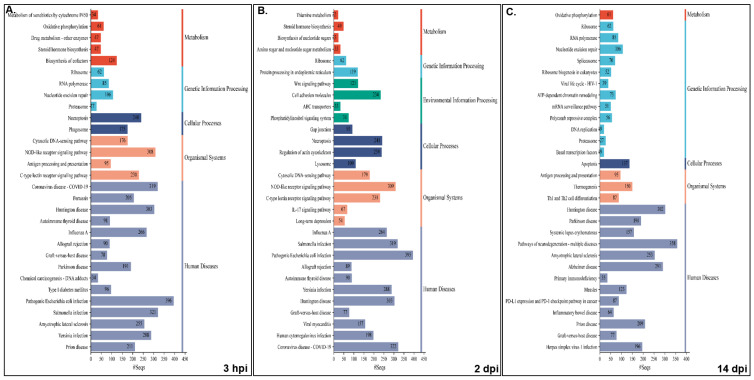
The top 30 enriched KEGG pathways in SI vs. SNI groups. (**A**) 3 hpi, (**B**) 2 dpi, and (**C**) 14 dpi. SI, susceptible infected fish; SNI, susceptible non-infected fish; hpi, hours post infection; dpi, days post infection.

**Figure 8 ijms-26-09220-f008:**
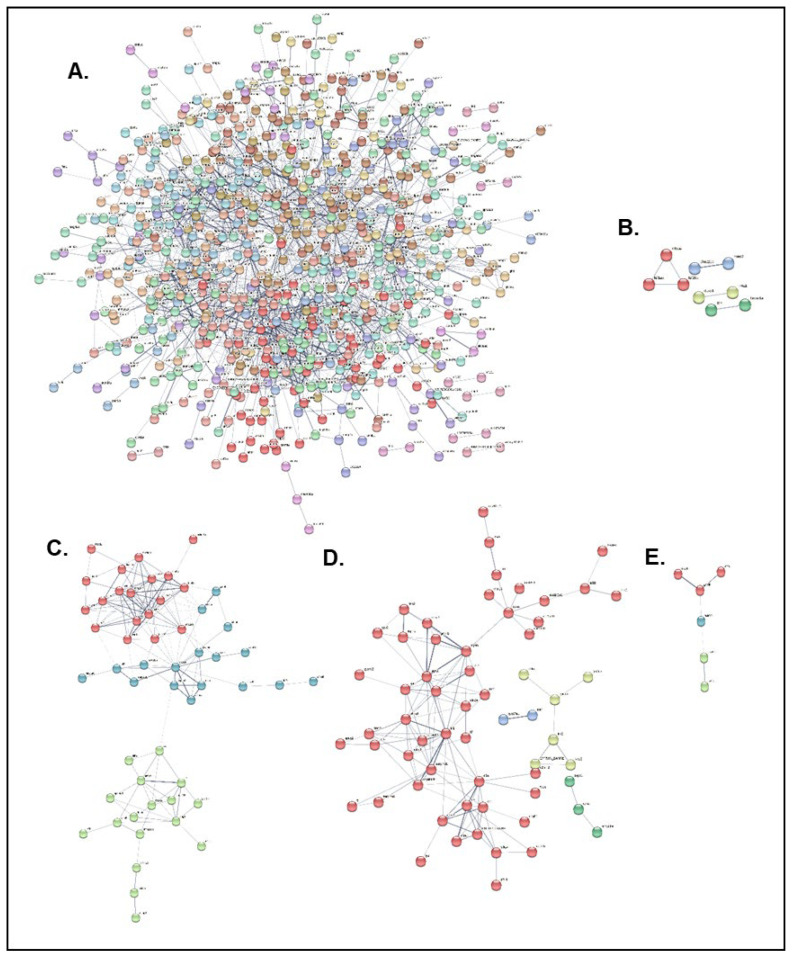
Protein–protein interaction (PPI) networks, with k-means clustering for the zebrafish orthologs of the differentially expressed genes in *D. labrax* (**A**) RI vs. RNI groups 3 hpi; (**B**) RI vs. RNI groups 2 hpi; (**C**) SI vs. SNI groups 3 hpi; (**D**) SI vs. SNI groups 2 dpi, and (**E**) SI vs. SNI groups 14 dpi. It is retrieved via API access to the STRING database (https://string-db.org) (accessed on 24 April 2025) and was performed based on the *Danio rerio* protein database. Each colored group represents a different cluster. The edges represent protein–protein interactions.

**Figure 9 ijms-26-09220-f009:**
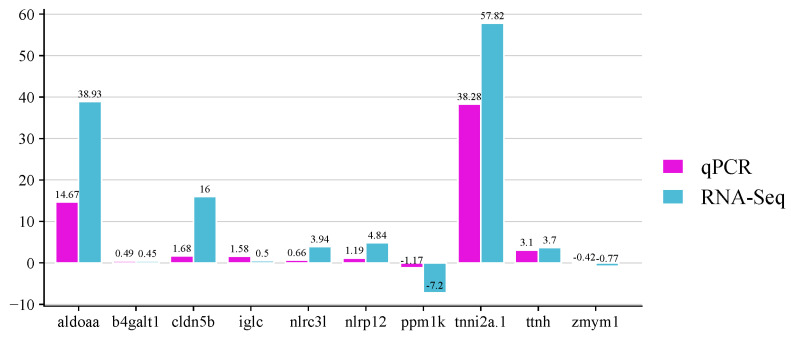
Validation of the RNA-Seq results by qPCR. Aldoaa, fructose-bisphosphate aldolase a; b4galt1, beta-galactosyltransferase 1-like; cldn5b, claudin-5-like; iglc, immunoglobulin light chain; nlrc3l, protein nlrc3-like; nlrp12, lrr and pyd domains-containing protein 12-like; ppm1k, protein phosphatase mitochondrial-like; tnni2a.1, troponin fast; ttnh, novel protein titin; zmym1, zinc finger mym-type protein 1-like.

**Table 1 ijms-26-09220-t001:** Enriched pathway categories in KEGG databases for RI vs. RNI groups; 3 hpi, 2 and 14 dpi.

Pathway Category	Enriched Pathways		
	3 hpi	2 dpi	14 dpi
Human Diseases	74	52	24
Genetic Information Processing	13	16	8
Metabolism	19	12	9
Organismal Systems	67	46	23
Cellular Processes	16	12	8
Environmental Information Processing	33	25	6

**Table 2 ijms-26-09220-t002:** Enriched pathway categories in the KEGG databases for SI vs. SNI groups; 3 hpi, 2 and 14 dpi.

Pathway Category	Enriched Pathways		
	3 hpi	2 dpi	14 dpi
Human Diseases	73	18	38
Genetic Information Processing	13	2	13
Metabolism	19	10	10
Organismal Systems	56	10	36
Cellular Processes	18	4	10
Environmental Information Processing	25	4	13

**Table 3 ijms-26-09220-t003:** Protein–protein interaction analysis of NNV-infected vs. –non-infected DEGs for resistant and susceptible experimental fish groups (RI, resistant infected fish; RNI, resistant non-infected fish; SI, susceptible infected fish; SNI, susceptible non-infected fish; hpi, hours post infection; dpi, days post infection).

DEGs	RI vs. RNI		SI vs. SNI	
	Initial	Final	Initial	Final
3 hpi	1898	871	169	84
2 dpi	90	57	237	116
14 dpi	-	-	22	18

**Table 4 ijms-26-09220-t004:** Top 10 hub genes with highest maximal clique centrality (MCC) score in PPI network of head kidney transcriptome data for resistant experimental fish groups, at each time point. The gene ID corresponds to Cytoscape gene references (RI, resistant infected fish; RNI, resistant non-infected fish; hpi, hours post infection; dpi, days post infection).

MCC Rank	Gene ID	Gene Name	Gene Symbol	logFold Change
**RI vs. RNI 3 hpi**				
**1**	7955.ENSDARP00000069524	Golgi SNAP receptor complex member 2 isoform X2	*gosr2*	2.97
**2**	7955.ENSDARP00000107557	CD59 glycoprotein	*cd59*	−6.88
**3**	7955.ENSDARP00000137308	protein transport protein Sec23A-like	*sec23a*	4.85
**4**	7955.ENSDARP00000121390	SCFD1 protein	*scfd1*	2.23
**5**	7955.ENSDARP00000149838	vesicle-trafficking protein SEC22c	*sec22c*	2.21
**6**	7955.ENSDARP00000103358	antithrombin-III	*serpina1*	−3.51
**7**	7955.ENSDARP00000007824	protein cornichon homolog 1	*cnih1*	2.20
**8**	7955.ENSDARP00000058106	multiple coagulation factor deficiency protein 2	*mcfd2*	4.16
**9**	7955.ENSDARP00000083123	vesicular integral-membrane protein VIP36	*lman2*	2.55
**10**	7955.ENSDARP00000023281	pre-mRNA 3′ end processing protein WDR33-like, partial	*wdr3*	2.82
**RI vs. RNI 2 dpi**				
**1**	7955.ENSDARP00000065336	kinesin-like protein KIF20A isoform X1	*kif20a*	2.29
**2**	7955.ENSDARP00000139080	kinesin family member 5Aa isoform X1	*kif5aa*	−5.75
**3**	7955.ENSDARP00000129632	kinesin-like protein KIF1A isoform X1	*kif1aa*	−6.22
**4**	7955.ENSDARP00000001911	nuclear receptor subfamily 4 group A member 1	*nr4a1*	−2.09
**5**	7955.ENSDARP00000137487	Dual specificity protein phosphatase 1	*dusp1*	−2.08
**6**	7955.ENSDARP00000022921	heart- and neural crest derivatives-expressed protein 2	*hand2*	−6.30
**7**	7955.ENSDARP00000130156	paired like homeobox 2Bb	*phox2bb*	−5.87
**8**	7955.ENSDARP00000038993	heme oxygenase-like	*hmox1a*	−2.62
**9**	7955.ENSDARP00000140751	interleukin-1 beta-like	*il1b*	−2.60

**Table 5 ijms-26-09220-t005:** Top 10 hub genes with highest maximal clique centrality (MCC) score in PPI network of head kidney transcriptome data for susceptible experimental fish groups, at each time point. The gene ID corresponds to Cytoscape gene references (SI, susceptible infected fish; SNI, susceptible non-infected fish; hpi, hours post infection; dpi, days post infection).

MCC Rank	Gene ID	Gene Name	Gene Symbol	logFold Change
**SI vs. SNI 3 hpi**				
**1**	7955.ENSDARP00000090306	Myosin-7 Myosin heavy chain 7	*myh7*	−11.65
**2**	7955.ENSDARP00000124963	troponin T, cardiac muscle isoforms isoform X10	*tnnt2a*	−9.42
**3**	7955.ENSDARP00000037759	troponin I, fast skeletal muscle-like	*tnni2a*.4	−9.59
**4**	7955.ENSDARP00000005224	alpha-actinin-3	*actn3a*	−3.93
**5**	7955.ENSDARP00000062369	Actin, alpha cardiac	*actc1a*	−4.69
**6**	7955.ENSDARP00000094172	troponin T, fast skeletal muscle isoforms-like	*tnnt3a*	−11.30
**7**	7955.ENSDARP00000018197	myosin, light polypeptide 3, skeletal muscle	*mylz3*	−8.60
**8**	7955.ENSDARP00000000488	Troponin I	*tnni1a*	−9.53
**9**	7955.ENSDARP00000099532	titin-like isoform X1	*ttn.2*	−8.27
**10**	7955.ENSDARP00000043931	Sarcoplasmic/endoplasmic reticulum calcium ATPase 1	*atp2a1*	−7.99
**SI vs. SNI 2 dpi**				
**1**	7955.ENSDARP00000037025	alpha-2-HS-glycoprotein	*ahsg1*	−7.09
**2**	7955.ENSDARP00000045815	plasminogen, partial	*plg*	−5.80
**3**	7955.ENSDARP00000121053	Uricase	*uox*	−8.30
**4**	7955.ENSDARP00000017257	alanine–glyoxylate and serine–pyruvate aminotransferase b	*agxtb*	−7.95
**5**	7955.ENSDARP00000017995	heparin cofactor 2	*serpind1*	−9.93
**6**	7955.ENSDARP00000068371	alanine–glyoxylate and serine–pyruvate aminotransferase a	*agxta*	−7.36
**7**	7955.ENSDARP00000108264	hyaluronan-binding protein 2-like	*habp2*	−8.74
**8**	7955.ENSDARP00000123643	alpha-2-antiplasmin	*serpinf2a*	−8.10
**9**	7955.ENSDARP00000115822	complement component C8 alpha chain	*c8a*	−8.84
**10**	7955.ENSDARP00000088095	complement C5	*c5*	−7.72
**SI vs. SNI 14 dpi**				
**1**	7955.ENSDARP00000063799	glyceraldehyde-3-phosphate dehydrogenase	*gapdh*	−9.53
**2**	7955.ENSDARP00000130083	actin, alpha cardiac muscle 1 isoform X4	*actc1b*	−4.19
**3**	7955.ENSDARP00000115698	ACTN2 protein	*actn2b*	−6.62
**4**	7955.ENSDARP00000054986	beta-actin	*actb1*	−7.77
**5**	7955.ENSDARP00000070225	aldolase a, fructose-bisphosphate, b	*aldob*	−11.27
**6**	7955.ENSDARP00000134589	ryanodine receptor 1-like isoform X2	*ryr2b*	−4.51

## Data Availability

The raw sequencing data are available in the NCBI database (accession no. PRJNA1030357).

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
