# Peer review of "Deciphering European Sea Bass (Dicentrarchus labrax) Resistance to Nervous Necrosis Virus by Transcriptome Analysis from Early Infection Towards Establishment of Virus Carrier State"

_ijms, 2025, doi:10.3390/ijms26189220_

Round 1
Reviewer 1 Report
Comments and Suggestions for Authors
It is a meaningful work to analyze the transcriptome of resistant and susceptible European sea bass after Nervous necrosis virus infection. The study work is well designed. The results and conclusion are well supported by the data.
But there is one limit of this study. Lots of pathways are found related with the Nervous necrosis virus. Most of them are very important for the life. Then, how do the author plan to do breeding based on so much important genes? To modify it, knock it out, or what other want to do for selective breeding. You konw no key gene could responsible for the virus infection base on the results. Then what to do next step?
Introduction It is well written to illustrate the object, and the scientific question. It is suggested to supply some limitations on the research work on Nervous necrosis virus, then to show the improvement and the innovations in this study.
Results, The definition on different groups is still not clear. How to identify RI and RNI, and what is the difference between SI and SNI? Given detail notes on Figure 1, what's the meaning of >50? The observation lasted to 28 dpi, why the transcriptom was only analyze at 14 dpi, not at 28 dpi?
Figure 2, why not compare the difference at 14 dpi between R and S groups?
Table 1, Table 2 and Table 3 should be three line table. The format of table is wrong.
Section 2.7, this is the results to prove the results of transcriptom. Therefore, it is suggested to put the Figure S2 into the paper to show the difference of different genes expression. Moreover, genes used to do gene expression should be use small letter and italic.
Conclusion It is suggested to illustrate the further study or practical plan or work in the future based on the conclusion. Could authors give some special pathway which could be consider in the breeding, not so much related genes?
Author Response
|
Response to Reviewer 1 Comments
|
||
|
1. Summary |
|
|
|
We would like to thank you very much for the time and effort to review this manuscript. Please find the detailed responses below and the corresponding updated information that was included to clarify our results highlighted in the re-submitted files. We believe that the clarifications/additions based on the reviewers’ comments have greatly improved the quality of our manuscript.
|
||
|
2. Questions for General Evaluation |
Reviewer’s Evaluation |
Response and Revisions |
|
Does the introduction provide sufficient background and include all relevant references? |
Can be improved |
Please check responses 2, and 3. |
|
Is the research design appropriate? |
Yes |
|
|
Are the methods adequately described? |
Yes |
|
|
Are the results clearly presented? |
Yes |
|
|
Are the conclusions supported by the results? |
Can be improved |
Please check responses 2, and 8. |
|
3. Point-by-point response to Comments and Suggestions for Authors |
||
|
Comments 1: It is a meaningful work to analyze the transcriptome of resistant and susceptible European sea bass after Nervous necrosis virus infection. The study work is well designed. The results and conclusion are well supported by the data. |
||
|
Response 1: We would like to thank the reviewer for the kind words on our work.
|
||
|
Comments 2: But there is one limit of this study. Lots of pathways are found related with the Nervous necrosis virus. Most of them are very important for the life. Then, how do the author plan to do breeding based on so much important genes? To modify it, knock it out, or what other want to do for selective breeding. You know no key gene could responsible for the virus infection base on the results. Then what to do next step? |
||
|
Response 2: We would like to thank the reviewer for that specific comment. We totally agree that the study limitation which is pointed out is very important. Being actively involved in the field of fish disease resistance, the fact that no key gene (or a limited number of genes), found in the present or independent studies, can be considered solely responsible for the resistance mechanism in order to be studied by knockouts and be easily applied in selective breeding, is also troubling us. For that reason, in order to gain a more holistic view for the mechanisms which are implicated in disease resistance we performed an extensive analysis on genes and pathways enrichment, and predicted protein-protein interactions in both resistant and susceptible families. For example, from our and other studies results it is clear that the pathways implicated in lipid metabolism are essential for conferring resistance to NNV. Our next step would be to further integrate our findings with other organ transcriptome analysis (e.g. brain, spleen, liver) based on RNA-seq data which are or will be publicly available, in order to find pathways implicated in sea bass resistance (in a ‘universal’ point of view). By combining these results with genomic and/or proteomic analysis we expect to find more specific key gene(s) and be able to apply this integrated knowledge to selective breeding. In our opinion, the present study is an essential step towards that direction, since we strongly believe that publication/combination of such studies offer a solid basis to move forward. In order to emphasize this point, the above mentioned plans have been added to the conclusion section (page 18 of revised manuscript), as it is also suggested by comments 8.
|
||
|
Comments 3: Introduction: It is well written to illustrate the object, and the scientific question. It is suggested to supply some limitations on the research work on Nervous necrosis virus, then to show the improvement and the innovations in this study. |
||
|
Response 3: Thank you for your suggestion. We agree with your comment. In our opinion, two important limitations in NNV resistance-related research are i. the fact that NNV resistance is a quantitative trait, therefore it is controlled by many factors, including gene expression variations, DNA methylation, histone modification, environmental factors and their interactions, and ii. there are no many studies on differences between symptomatic and asymptomatic virus carriers. Both facts are mentioned in introduction (lines 90-103 of the submitted manuscript) but it is not emphasized enough that our study follows the approach of transcriptome analysis of two genetically distinct families in early and later time point in order to improve results regarding these two limitations. Therefore, we have added the following in the introduction: “The differences in the two families’ responses indicate the essential elements which result in disease resistance and provide information regarding the NNV resistance trait which are not covered solely by genetic analysis. Moreover, analysis of a time point (14 dpi) when fish appear to act as asymptomatic carriers also, revels specific host mechanisms that can be key parts of disease resistance.”
|
||
|
Comments 4: Results: The definition on different groups is still not clear. How to identify RI and RNI, and what is the difference between SI and SNI? Given detail notes on Figure 1, what's the meaning of >50? The observation lasted to 28 dpi, why the transcriptome was only analyze at 14 dpi, not at 28 dpi? |
||
|
Response 4: Thank you for pointing out these issues which need further clarification. Throughout the manuscript, the letter R is used to symbolize the resistant family and the letter S is used to symbolize the susceptible family. The letter I following R or S is used for the infected groups and NI is used for the non-infected groups. Therefore, as stated in the legend of figure 1: RI is the symbol for resistant infected fish, RNI symbolizes resistant non-infected fish, SI stands for susceptible infected fish and SNI symbolizes the susceptible non-infected fish. In order to facilitate the groups’ identification, these definition were added in section 2.1 (page 3 of revised manuscript). As described in section 4.3 (page 19, lines 1234-1235 of the original submission pdf), fish were allocated in groups of 20 and 50 for family R and family S, respectively. The required numbers of fish were calculated based on a preliminary mortality experiment. The symbol > was used by mistake instead of =, and it is removed in the revised figure. And we also noted that by mistake the RI fish number was 50 while it should be 20, while the SNI was 20 while it should be 50. These mistakes were corrected in the revised figure 1. We regret the errors. Finally, as described in discussion (page 5, lines 541-553 of the original submission pdf), the time points were selected based on our previous study [ref. 36. Toubanaki, D.K.; Efstathiou, A.; Tzortzatos, O.P.; Valsamidis, M.A.; Papaharisis, L.; Bakopoulos, V.; Karagouni, E. Nervous Necrosis Virus Modulation of European Sea Bass (Dicentrarchus labrax, L.) Immune Genes and Transcriptome towards Establishment of Virus Carrier State. Int. J. Mol. Sci. 2023, 24, 16613. https://doi.org/10.3390/ijms242316613], where it was found that 14 dpi seems to be a key time point for establishment of a virus carrier state in the host, since in the 14 dpi time point both families had a considerable amount of the replicated virus in brain, with similar levels, but the fish didn’t have any clinical signs of the disease. For that reason, we chose to study the transcriptome on 14 dpi and not at the final time point (28 dpi).
|
||
|
Comments 5: Figure 2, why not compare the difference at 14 dpi between R and S groups? |
||
|
Response 5: Thank you for pointing this out. We have compared the difference between R and S group 14 dpi, and no statistically significant difference was found.
|
||
|
Comments 6: Table 1, Table 2 and Table 3 should be three line table. The format of table is wrong. |
||
|
Response 6: Thank you for pointing this out. We really appreciate this comment. Therefore, we have converted Tables 1, 2 and 3 to three line table format, to the best of our understanding.
|
||
|
Comments 7: Section 2.7, this is the results to prove the results of transcriptom. Therefore, it is suggested to put the Figure S2 into the paper to show the difference of different genes expression. Moreover, genes used to do gene expression should be use small letter and italic. |
||
|
Response 7: Thank you for pointing this out. We agree with this comment. Therefore, we have added figure S2 to the manuscript as figure 9. We also converted all genes used for gene expression to small letter and italic format.
|
||
|
Comments 8: Conclusion: It is suggested to illustrate the further study or practical plan or work in the future based on the conclusion. Could authors give some special pathway which could be consider in the breeding, not so much related genes? |
||
|
Response 8: We would like to thank the reviewer for that specific comment. As analyzed in response 2, our next step would be to further integrate our findings with other organ transcriptome analysis (e.g. brain, spleen, liver) based on RNA-seq data which are or will be publicly available, in order to find pathways implicated in sea bass resistance (in a ‘universal’ point of view). By combining these results with genomic and/or proteomic analysis we expect to find more specific key gene(s) and be able to apply this integrated knowledge to selective breeding. In our opinion, the present study is an essential step towards that direction, since we strongly believe that publication/combination of such studies offer a solid basis to move forward. In order to emphasize this point, the following text had been added to the conclusion section (page 18 of revised manuscript) “Our future plans are focused on further integration of the present findings with other organ transcriptome analysis (e.g. brain, spleen, liver) based on RNA-seq data which are or will be publicly available, in order to find pathways implicated in sea bass resistance (in a ‘universal’ point of view). For example, from our and other studies results it seems that not only immune related, but also motor protein or lipid metabolism related pathways are essential for conferring resistance to NNV. By combining these results with genomic and/or proteomic analysis we expect to find more specific key gene(s) and be able to apply this integrated knowledge to selective breeding.”
|
||
|
4. Response to Comments on the Quality of English Language |
||
|
Point 1: The English is fine and does not require any improvement |
||
|
Response 1: We thank the reviewer for the kind words. |
||

Reviewer 2 Report
Comments and Suggestions for Authors
Elucidating the basiss of disease susceptibility and resistance in fish has been one major goal in aquaculture research. In this work, the authors present a transcriptome/RNAseq analysis of differential gene expression between two sea bass populations: resistant and susceptible to the nodavirus NNV. As expected from a study of this type a great deal of data were collected that are thoroughly analyzed through the manuscript. In my opinion, though, the discussion section is overextended, providing detailed information about dozens of genes. I believe the focus should have been on the main factors playing a role in resistant to viral diseases (such as innate and adaptive immune responses and other pathways related to defense response to viruses), while the discussion of other biological processes could have been more concise, unless actually relevant to resistance to disease. On this regard, it is surprising that a number of genes involved in the innate immune response previously reported in other transcriptome/RNAseq studies (i.e: interferon regulatory factors, mx, viperin) are not mentioned in this work.
In addition, I would advise to draw a scheme with the more relevant genes/pathways that came up after the transcriptomic analysis. I am aware of the difficulty to pinpoint specific genes as responsible of resistance to viral infection, but after reading eleven pages of Discussion I wasn´t sure yet to be able to discern what makes a fish resistant to NNV pathogenesis.
Specific comments
1.- In figure 2, I wonder why not plotting together susceptible and resistant fish to better compare viral loads.
2.- Resistant sea bass infected with NNV have significant levels of viral RNA in brain. Will those fish clear the virus eventually, or they will remain virus carriers for life?
3.- I don´t quite understand why the qRT-PCR data are presented as a supplementary figure. qPCR analysis usually provides a strong support to transcriptomic data. As mentioned earlier, it is odd that not a single innate immune gene was selected for qPCR analysis. One interesting gene to be checked by qPCR is il1β that was found down-regulated in resistant fish.
A few references on transcriptomic/RNAseq studies on response to viral infection in fish seem to be missing. One relevant work that should be cited: Polinski MP, Bradshaw JC, Rise ML, Johnson SC, Garver KA. Sockeye salmon demonstrate robust yet distinct transcriptomic kidney responses to rhabdovirus (IHNV) exposure and infection. Fish Shellfish Immunol. 2019 Nov;94:525-538.
Author Response
|
Response to Reviewer 2 Comments
|
||
|
1. Summary |
|
|
|
We would like to thank you very much for the time and effort to review this manuscript. Please find the detailed responses below and the corresponding updated information that was included to clarify our results highlighted in the re-submitted files. We believe that the clarifications/additions based on the reviewers’ comments have greatly improved the quality of our manuscript.
|
||
|
2. Questions for General Evaluation |
Reviewer’s Evaluation |
Response and Revisions |
|
Does the introduction provide sufficient background and include all relevant references? |
Yes |
|
|
Is the research design appropriate? |
Yes |
|
|
Are the methods adequately described? |
Yes |
|
|
Are the results clearly presented? |
Can be improved |
Please check response 4. |
|
Are the conclusions supported by the results? |
Can be improved |
Please check responses 5, and 6. |
|
3. Point-by-point response to Comments and Suggestions for Authors |
||
|
Comments 1: Elucidating the basiss of disease susceptibility and resistance in fish has been one major goal in aquaculture research. In this work, the authors present a transcriptome/RNAseq analysis of differential gene expression between two sea bass populations: resistant and susceptible to the nodavirus NNV. As expected from a study of this type a great deal of data were collected that are thoroughly analyzed through the manuscript. |
||
|
Response 1: We would like to thank the reviewer for the kind words on our work.
|
||
|
Comments 2: In my opinion, though, the discussion section is overextended, providing detailed information about dozens of genes. I believe the focus should have been on the main factors playing a role in resistant to viral diseases (such as innate and adaptive immune responses and other pathways related to defense response to viruses), while the discussion of other biological processes could have been more concise, unless actually relevant to resistance to disease. On this regard, it is surprising that a number of genes involved in the innate immune response previously reported in other transcriptome/RNAseq studies (i.e: interferon regulatory factors, mx, viperin) are not mentioned in this work. |
||
|
Response 2: Thank you for that specific comment. It is an interesting point that has also troubled us during our results analysis and presentation. It is widely known that such transcriptomic studies result in a great deal of data, as you also mention, and it is really hard to present all these data in a single article. For that reason, most of the relevant literature is focused on immune response and specific defense pathways, as you also suggest. But research in this field still struggles to provide evidence for specific factors contributing to resistance despite a large amount of related data. For that reason, we chose to follow an alternative point of view on the discussion of our results, by presenting most of the implicated pathways and genes, and not focusing on some categories only. We strongly believe that a more ‘holistic’ view of the resistance mechanism will provide substantial information to be actually used in the field. For example, we see that motor proteins play a really important role in resistance mechanisms which in combination with immune factors may provide solution for virus control. We also noticed the fact that genes involved in innate immune response doesn’t seem to be in first line in the present work, and we plan to perform an extra analysis on that aspect of resistance in the near future.
|
||
|
Comments 3: In addition, I would advise to draw a scheme with the more relevant genes/pathways that came up after the transcriptomic analysis. I am aware of the difficulty to pinpoint specific genes as responsible of resistance to viral infection, but after reading eleven pages of Discussion I wasn´t sure yet to be able to discern what makes a fish resistant to NNV pathogenesis. |
||
|
Response 3: Thank you for pointing this out. Even though, we agree with your comment, the construction of a figure illustrating the more resistant relevant genes/pathways is above our current capabilities due to the high number of the resulting factors and the complexity of the pathways. We regret but we cannot provide the requested figure. However, we are trying to gather resources to be able to provide one such figure after integrating the present findings with other organ transcriptome analysis (e.g. brain, spleen, liver) based on RNA-seq data which are or will be publicly available, in order to find pathways implicated in sea bass resistance (in a ‘universal’ point of view).
|
||
|
Comments 4: 1.- In figure 2, I wonder why not plotting together susceptible and resistant fish to better compare viral loads. |
||
|
Response 4: Thank you for your suggestion. We have, accordingly, modified figure 2 in the revised manuscript.
|
||
|
Comments 5: 2.- Resistant sea bass infected with NNV have significant levels of viral RNA in brain. Will those fish clear the virus eventually, or they will remain virus carriers for life? |
||
|
Response 5: Thank you for pointing this out. In our opinion this comment is a really interesting scientific question, which intrigues us to analyze the ‘late’ time points e.g. 14 dpi. In a related study [ref. 36. Toubanaki, D.K.; Efstathiou, A.; Tzortzatos, O.P.; Valsamidis, M.A.; Papaharisis, L.; Bakopoulos, V.; Karagouni, E. Nervous Necrosis Virus Modulation of European Sea Bass (Dicentrarchus labrax, L.) Immune Genes and Transcriptome towards Establishment of Virus Carrier State. Int. J. Mol. Sci. 2023, 24, 16613. https://doi.org/10.3390/ijms242316613], it was found that viral RNA in brain remains high up to 28 dpi while mortality rates and clinical symptoms were minimized. It was also surprising that on 14 dpi an enhanced immune activity was recorded when specific immune related genes were analyzed. To the best of our knowledge, there was no previous publication, describing the transcriptome profile at these later time points (i.e. 14 and 28 dpi), of either NNV infected or asymptomatic sea bass. Unfortunately, our experimentation was terminated at 28 dpi time point, where viral load was still present in surviving fish brain. Therefore we didn’t have the chance to test whether NNV was eventually cleared from sea bass brain, or remained for the rest of fish life. Currently, we seeking funding in order to perform that interesting study, which will be great for understanding NNV action to fish brain. But at the present we cannot provide an answer to this question.
|
||
|
Comments 6: 3.- I don´t quite understand why the qRT-PCR data are presented as a supplementary figure. qPCR analysis usually provides a strong support to transcriptomic data. As mentioned earlier, it is odd that not a single innate immune gene was selected for qPCR analysis. One interesting gene to be checked by qPCR is il1β that was found down-regulated in resistant fish. |
||
|
Response 6: Thank you for pointing this out. We agree with this comment. Therefore, we have added figure S2 to the manuscript as figure 9. We also thank you for your comment regarding the selection of the genes for qPCR analysis. These genes were selected in relation with the hub genes results from the PPI analysis, instead of choosing genes belonging to specific pathways, in an effort to validate genes from various pathways. Interleukin 1b, was extensively analyzed in our previously published study [ref. 36], where an NNV resistant sea bass family was experimentally infected and specific immune related genes (including Il-1b) were analyzed in a time course study with many time points. In that study, Il-1b was also down-regulated 2 dpi, further confirming the transcriptome results.
|
||
|
Comments 7: A few references on transcriptomic/RNAseq studies on response to viral infection in fish seem to be missing. One relevant work that should be cited: Polinski MP, Bradshaw JC, Rise ML, Johnson SC, Garver KA. Sockeye salmon demonstrate robust yet distinct transcriptomic kidney responses to rhabdovirus(IHNV) exposure and infection. Fish Shellfish Immunol. 2019 Nov;94:525-538. |
||
|
Response 7: We thank the reviewer for the kind suggestion on viral infection and transcriptome head-kidney analysis. The suggested reference was added in the introduction section.
|
||
|
4. Response to Comments on the Quality of English Language |
||
|
Point 1: The English is fine and does not require any improvement |
||
|
Response 1: We thank the reviewer for the kind words. |
||

Reviewer 3 Report
Comments and Suggestions for Authors
The authors did RNA-seq analysis for resistant and susceptible European sea bass to nervous necrosis virus at three time points. The differentially expressed genes were identified, and functional gene analysis was done to reveal the molecular mechanisms of different groups of fish responding to the viral infection. This study is important and may be used to develop strategies to control viral diseases in the aquaculture industry. To improve the quality and readability of the manuscript, the following comments need to be addressed:
- Totally, how many tanks were used in the experiment, and the number of fish in each tank?
- NCBI accession No. PRJNA1030357 has only 9 samples: NNV infected (n=5) and non-infected (n=4) fish were subjected to sequencing, which are not consistent with the 45 libraries sequenced.
- How was the viral titer quantified?
- What is the reference genome used in the alignment.
- As only 2-5 samples in each group were used in the study, the results of PPI network constructed by Cytoscape were questionable.
- In the discussion section, some paragraphs, such as the fourth paragraph, are just a description statement.
- How were the top GOs or pathways determined.
- Lines 238-239: “The number of common significant down-regulated and up-regulated genes between each comparison are shown in the space between their volcano plots.” It is hard to understand the author’s description. In the middle of the plot, the dots were not selected as DETs based on the criteria defined in the manuscript.
- Line 1302: Please read the program manual to make sure Benjamini-Hochberg procedure was not used for multiple testing corrections (FDR).
- 1280-1282: How DESeq2 can remove the adapters?
- The supplementary figures couldn’t be found.
- Line 77: Leading ?
- Line 105: …”have gained attention ?? the last few”
- Lines 129, and 1351: focused on
- Ine 218: de-regulation
- Lines 246-251, 382: Grammar error.
Author Response
|
Response to Reviewer 3 Comments
|
||
|
1. Summary |
|
|
|
We would like to thank you very much for the time and effort to review this manuscript. Please find the detailed responses below and the corresponding updated information that was included to clarify our results highlighted in the re-submitted files. We believe that the clarifications/additions based on the reviewers’ comments have greatly improved the quality of our manuscript.
|
||
|
2. Questions for General Evaluation |
Reviewer’s Evaluation |
Response and Revisions |
|
Does the introduction provide sufficient background and include all relevant references? |
Can be improved |
Please check responses 13 - 15. |
|
Is the research design appropriate? |
Yes |
|
|
Are the methods adequately described? |
Can be improved |
Please check responses 2 – 5, 8, 11. |
|
Are the results clearly presented? |
Can be improved |
Please check responses 7, 9. |
|
Are the conclusions supported by the results? |
Can be improved |
Please check responses 6, 17. |
|
3. Point-by-point response to Comments and Suggestions for Authors |
||
|
Comments 1: The authors did RNA-seq analysis for resistant and susceptible European sea bass to nervous necrosis virus at three time points. The differentially expressed genes were identified, and functional gene analysis was done to reveal the molecular mechanisms of different groups of fish responding to the viral infection. This study is important and may be used to develop strategies to control viral diseases in the aquaculture industry. To improve the quality and readability of the manuscript, the following comments need to be addressed: |
||
|
Response 1: We would like to thank the reviewer for the kind words on our work.
|
||
|
Comments 2: 1. Totally, how many tanks were used in the experiment, and the number of fish in each tank? |
||
|
Response 2: Thank you for pointing out these issues which need further clarification. As described in section 4.3, “fish were stocked in an allocated tank system (in triplicate per family/per condition), in groups of 20 and 50 for family R and family S, respectively”. Therefore, 12 tanks were used in total. Each tank containing resistant fish had 20 fish while each tank containing susceptible fish had 50 fish.
|
||
|
Comments 3: 2. NCBI accession No. PRJNA1030357 has only 9 samples: NNV infected (n=5) and non-infected (n=4) fish were subjected to sequencing, which are not consistent with the 45 libraries sequenced. |
||
|
Response 3: Thank you for pointing this out. As mentioned in our cover letter “all sequencing raw data reads will be available in the NCBI database (Accession No. PRJNA1030357) until the review process is completed, however due to technical problems the process has not be completed at this moment for the 3 hpi and 2 dpi datasets. We hope that in the next few days we will be able to complete the data uploading. In any case, all data are at your disposal upon request”. Due to some difficulties, we are still trying to resolve the issue, which we hope will be resolved in the next 2-3 days, and for sure until the publication date if the manuscript will be accepted . We regret, the fact that our claims seem inconsistence but the issue was above our technical level.
|
||
|
Comments 4: 3. How was the viral titer quantified? |
||
|
Response 4: Thank you for pointing out these issue which need further clarification. As described in section 4.2, the NNV viral titer was performed by the end-point titration method. Serial dilutions of viral suspensions in EMEM - 10% FBS, were added in a 96-well plate seeded with monolayers of SSN-1 cells and incubated at 26 °C for 6 days. During this period, the cell monolayers were observed for the appearance of cytopathic effect (CPE) and the final titer, expressed as TCID50·mL−1, was estimated, as described by Reed LJ, Muench H. A Simple Method of Estimating Fifty Per Cent Endpoints. Am J Epidemiol. (1938) 27(3):493-497. doi: 10.1093/oxfordjournals.aje.a118408. The reference was added in the text.
|
||
|
Comments 5: 4. What is the reference genome used in the alignment. |
||
|
Response 5: Thank you for pointing out these issue which need further clarification. As described in section 4.6, “the reads were aligned to Dicentrarchus labrax reference genome (sea-bass_V1.0 - GCA_000689215.1) [30. Tine, M.; Kuhl, H.; Gagnaire, P.A.; Louro, B.; Desmarais, E.; Martins, R.S.; Hecht, J.; Knaust, F.; Belkhir, K.; Klages, S.; et al. European sea bass genome and its variation provide insights into adaptation to euryhalinity and speciation. Nat. Commun. 2014, 5, 5770. https://doi.org/10.1038/ncomms6770.]”
|
||
|
Comments 6: 5. As only 2-5 samples in each group were used in the study, the results of PPI network constructed by Cytoscape were questionable. |
||
|
Response 6: Thank you for pointing this out. We have also considered this issue. For that reason, we state in section 2.6, that the PPI analysis is used to “gain further insights into the disease resistance mechanisms, by manually curating all initially found DEGs to find zebrafish homologs”. No results from PPI analysis are definite, we only gain indications to perform further functional analysis. Our approach was based on similar independent studies (e.g. 18. Pereiro, P.; Figueras, A.; Novoa, B. RNA-Seq analysis of juvenile gilthead sea bream (Sparus aurata) provides some clues regarding their resistance to the nodavirus RGNNV genotype. Fish Shellfish Immunol. 2023, 134, 108588. doi: 10.1016/j.fsi.2023.108588; 23. Wang, L.; Xu, X.; Zhang, Z.; Li, K.; Yang, Y.; Zheng, W.; Sun, H.; Chen, S. Transcriptome analysis and pro-tein-protein interaction in resistant and susceptible families of Japanese flounder (Paralichthys olivaceus) to un-derstand the mechanism against Edwardsiella tarda. Fish Shellfish Immunol. 2022, 123, 265–228. https://doi.org/10.1016/j.fsi.2022.02.055.)
|
||
|
Comments 7: 6. In the discussion section, some paragraphs, such as the fourth paragraph, are just a description statement. |
||
|
Response 7: Thank you for pointing this out. We have indeed use descriptive statements in the discussion as paragraph 4, in order to facilitate the reader to summarize all different aspects of elements conferring disease resistance to a host organism and then to be able to read each element in more detail. If the editor agrees that this paragraph should be removed, we can remove it.
|
||
|
Comments 8: 7. How were the top GOs or pathways determined. |
||
|
Response 8: Thank you for pointing out these issues which need further clarification. As described in section 4.7 “The annotated DEGs were subjected to Fisher’s exact test and were considered significantly enriched in GO terms when their Bonferroni adjusted p-value was less than 0.05. The enriched KEGG pathways were determined by Gene Set Enrichment Analysis (GSEA), with ad-justed p-value less than 0.05”.
|
||
|
Comments 9: 8. Lines 238-239: “The number of common significant down-regulated and up-regulated genes between each comparison are shown in the space between their volcano plots.” It is hard to understand the author’s description. In the middle of the plot, the dots were not selected as DETs based on the criteria defined in the manuscript. |
||
|
Response 9: Thank you for pointing out this issue which needs further clarification. We agree that the figure legend contains a confusing description. Therefore, we re-phrase as follows: “The number of common significant down-regulated and up-regulated genes between the comparison of the two families at each time point are shown in the middle of the figure, i.e. below the RI vs RNI and above the SI vs SNI volcano plots.”
|
||
|
Comments 10: 9. Line 1302: Please read the program manual to make sure Benjamini-Hochberg procedure was not used for multiple testing corrections (FDR). |
||
|
Response 10: Could you please clarify that specific comment? As described in the text we used the Bonferroni adjusted p-value, as proposed by many publications in the field. Please let us know if we need to change something.
|
||
|
Comments 11: 10. 1280-1282: How DESeq2 can remove the adapters? |
||
|
Response 11: Thank you for pointing this out. The adapters’ removal software was omitted by mistake. Therefore, we have revised the 1280-1282 sentence as follows: “Raw data were processed to remove adapters by Trimmomatic, and were normalized for inherent systematic or experimental biases, using the Bioconductor package DESeq2”. We regret the error.
|
||
|
Comments 12: 11. The supplementary figures couldn’t be found. |
||
|
Response 12: The supplementary figures were submitted as part of the Toubanaki et al._IJMS_manuscript-SI pdf file and they were not submitted separately. We will submit the supplementary figure separately in the revised material of the manuscript in order to be found more easily. We are sorry for the inconvenience.
|
||
|
Comments 13: 12. Line 77: Leading ? |
||
|
Response 13: Thank you for pointing this out. The word leading was left by mistake and it is removed in the revised manuscript.
|
||
|
Comments 14: 13. Line 105: …”have gained attention ?? the last few” |
||
|
Response 14: Thank you for pointing this out. We agree that this phrase isn’t optimal. Therefore, we have re-phrased it as follows: “the use of genetically characterized pathogen resistant and susceptible hosts for transcriptome analysis have gained attention been the subject of increasing investigation the last few years decade”.
|
||
|
Comments 15: 14. Lines 129, and 1351: focused on |
||
|
Response 15: Thank you for pointing this out. We agree with your correction and regret the errors. Therefore, we have corrected focused in to focused on, in both lines.
|
||
|
Comments 16: 15. Ine 218: de-regulation |
||
|
Response 16: Thank you for pointing this out. We agree with this correction and regret the error. Therefore, we have corrected de-regulation to deregulation
|
||
|
Comments 17: 16. Lines 246-251, 382: Grammar error. |
||
|
Response 17: Thank you for pointing this out. We agree with these comments and regret the errors. Therefore, we have corrected: - the 246-251 paragraph as follows: “In order to identify the potential molecular mechanisms underlying VNN disease resistance, we followed a three step strategy, which consisted of i. identification of DEGs resulting from the comparison of infected versus non-infected groups for each family at each time point and their GO classification; ii. identification of family-specific GOs of the up- and down-regulated genes for the resistant and susceptible families at each time point, separately; and iii. identification of family-specific GOs at all-time points” - the 382 sentence as follows: “Pathway analysis was performed for NNV infected versus non-infected…”
|
||
|
4. Response to Comments on the Quality of English Language |
||
|
Point 1: The English could be improved to more clearly express the research. |
||
|
Response 1: We thank the reviewer for the suggestion. Following the incorporation of all suggested corrections by the reviewers, the manuscript was proof-read by an English native speaker. We hope that the revised manuscript has been adequately improved. |
||

Round 2
Reviewer 2 Report
Comments and Suggestions for Authors
No further comments.